# GPRChinaTemp1km: a high-resolution monthly air temperature dataset for China (1951–2020) based on machine learning

Qian He[1, 2], Ming Wang[1, 3], Kai Liu[1, 3], Kaiwen Li[1, 2], Ziyu Jiang[1, 2]

[1] Academy of Disaster Reduction and Emergency Management, Beijing Normal University, 100875 Beijing, China
[2] Faculty of Geographical Science, Beijing Normal University, 100875 Beijing, China
[3] School of National Safety and Emergency Management, Beijing Normal University, 100875 Beijing, China

*Correspondence to*: Ming Wang (wangming@bnu.edu.cn)

**Abstract**

An accurate spatially continuous air temperature dataset is crucial for multiple applications in environmental and ecological sciences. Existing spatial interpolation methods have relatively low accuracy and the resolution of available long-term gridded products of air temperature for China is coarse. Point observations from meteorological stations can provide long-term air temperature data series but cannot represent spatially continuous information. Here, we devised a method for spatial interpolation of air temperature data from meteorological stations based on powerful machine learning tools. First, to determine the optimal method for interpolation of air temperature data, we employed three machine learning models: random forest, support vector machine, and Gaussian process regression. Comparison of the mean absolute error, root mean square error, coefficient of determination, and residuals revealed that Gaussian process regression had high accuracy and clearly outperformed the other two models regarding interpolation of monthly maximum, minimum, and mean air temperatures. The machine learning methods were compared with three traditional methods used frequently for spatial interpolation: inverse distance weighting, ordinary kriging, and ANUSPLIN (short for Australian National University Spline). Results showed that the Gaussian process regression model had higher accuracy and greater robustness than the traditional methods regarding interpolation of monthly maximum, minimum, and mean air temperatures in each month. Comparison with the TerraClimate (Monthly Climate and Climatic Water Balance for Global Terrestrial Surfaces), FLDAS (Famine Early Warning Systems Network (FEWS NET) Land Data Assimilation System), and ERA5 (ECMWF Climate Reanalysis) datasets revealed that the accuracy of the temperature data generated using the Gaussian process regression model was higher. Finally, using the Gaussian process regression method, we produced a long-term (January 1951 to December 2020) gridded monthly air temperature dataset with 1 km resolution and high accuracy for China, which we named GPRChinaTemp1km. The dataset consists of three variables: monthly mean air temperature, monthly maximum air temperature, and monthly minimum air temperature. The obtained GPRChinaTemp1km data were used to analyse the spatiotemporal variations of air temperature using Theil–Sen median trend analysis in combination with the Mann–Kendall test. It was found that the monthly mean and minimum air temperatures across China were characterized by a significant trend of increase in each month, whereas monthly maximum air temperature showed a more spatially heterogeneous pattern with significant increase, non-significant increase,

and non-significant decrease. The GPRChinaTemp1km dataset is publicly available at https://doi.org/10.5281/zenodo.5112122 (He et al., 2021a) for monthly maximum air temperature, at https://doi.org/10.5281/zenodo.5111989 (He et al., 2021b) for monthly mean air temperature and at https://doi.org/10.5281/zenodo.5112232 (He et al., 2021c) for monthly minimum air temperature.

## 1 Introduction

Air temperature is a fundamental variable in various research fields that include the impact of global warming and climate change, ecology, hydrology, agriculture, and human health (Sippel et al., 2020; Abatzoglou et al., 2018; Pathak et al., 2018; Chen et al., 2018). The monthly temperature data is crucial for multiple studies and applications such as agriculture (Meshram et al., 2020), meteorological disasters (Tigkas et al., 2019) and ecology (Leihy et al., 2018). Long-term records of air temperature data with high spatial resolution are necessary for such research. Generally, air temperature data are measured by meteorological station networks or simulated using numerical climate models (dos Santos, 2020; Fu and Weng, 2018). Meteorological stations can provide long-term point-based information on observed air temperature; however, they cannot reflect spatially continuous information regarding regional air temperature. The downscaling technique is often used to obtain the high-resolution dataset using coarse-resolution products, while there are multiple low spatial resolution datasets, such as the Climatic Research Unit (CRU) (Harris et al., 2014), the Global Precipitation Climatology Centre (GPCC) (Schneider et al., 2014; Becker et al., 2013), and Willmott & Matsuura (W&M) (Matsuura and Willmott, 2012), are generated using the data from the observational stations. Interpolation is a reliable way to produce spatial continuous datasets using the observed station data (Peng et al., 2019). The reanalysis based data usually have low spatial resolution, which limits their ability to reflect the effects of complex topographies, land surface characteristics, and other processes on climate systems (Peng et al., 2019; Xu et al., 2017). Besides, some reanalysis products have uncertainty per se (Tang et al., 2020b; Yin et al., 2021) .

Various interpolation techniques that include inverse distance weighting (IDW) and ordinary kriging (OK) (Dawood, 2017; Li et al., 2011a, 2012; Hadi and Tombul, 2018; Stahl et al., 2006; Benavides et al., 2007; Duhan et al., 2013) are often employed to derive gridded temperature datasets for data-sparse areas. However, the accuracy of the derived results depends on the density of the meteorological stations used for the interpolation (Wang et al., 2017; Peng et al., 2019; Gao et al., 2018; Peng et al., 2014). Using conventional methods for data interpolation in areas with uneven coverage of meteorological stations could diminish the accuracy of the derived data (dos Santos, 2020; Li et al., 2018). The network of meteorological stations in China is characterized by irregular spatial coverage. For example, the observation network has low density in mountain areas (Gao et al., 2018; dos Santos, 2020; Guo et al., 2020), especially on the Tibetan Plateau (Xu et al., 2018; Zhang et al., 2016). Additionally, the number of meteorological stations operational in China in the 1950s was low. Therefore, use of conventional interpolation methods cannot guarantee the accuracy of the derived spatial datasets of air temperature across China. Although various air temperature products are available, e.g., the TerraClimate (Abatzoglou et al., 2018), FLDAS (McNally et al., 2017), and ERA5 (Copernicus Climate Change Service (C3S), 2017) datasets, their spatial resolution is usually coarse (2.5 arc minutes,

0.1 arc degrees, and 0.25 arc degrees, respectively), which restricts their ability to reflect the topographical characteristics and spatial heterogeneity of air temperature across China (Peng et al., 2019; Zhang et al., 2016). Thus, demand remains for long-term spatially continuous dataset of air temperature with a high spatial resolution.

In comparison with traditional interpolation techniques, machine learning methods are better able to model nonlinear and highly interactive relationships (Xu et al., 2018). Using mud content samples from the southwest margin of Australia, Li et al. (2011a) proved the superior performance of machine learning methods in application to spatial interpolation of environmental variables. Subsequent application of machine learning methods further confirmed their effectiveness as tools for interpolation of environmental variables, in which secondary information considered such as slope, latitude and longitude can improve the performance of machine learning (Li et al., 2011b, Appelhans et al., 2015; Zhu et al., 2018; Alizamir et al., 2020; Kisi et al., 2017). Many previous studies have demonstrated the potential of machine learning techniques in application to estimation of air temperature in small regions, although most such studies interpolated air temperature using satellite-derived predictors such as the Land Surface Temperature and Normalised Difference Vegetation Index based on MODIS products (Appelhans et al., 2015; dos Santos, 2020; Meyer et al., 2016; Xu et al., 2018; Zhang et al., 2016; Yoo et al., 2018). However, MODIS data are only available from 2000, which means that air temperature in earlier years cannot be interpolated using such products. Moreover, optical remote sensing images are easily affected by clouds, limiting the ability of associated models to produce long-term spatially continuous datasets for air temperature across large regions such as China (Dong and Xiao, 2016; Mao et al., 2019; Xiao et al., 2018, p.2013–2016). Therefore, it is necessary to develop a universal model to interpolate long-term air temperature datasets for China. However, how best to design a simple and accurate model for temperature interpolation using machine learning remains unclear.

To interpolate air temperature across China, we employed three machine learning approaches: random forest (RF), support vector machine (SVM), and Gaussian process regression (GPR). Both RF and SVM have been proven effective in previous studies on remote-sensing-based air temperature estimation studies (Yoo et al., 2018; Zhang et al., 2016; Ho et al., 2014; Zeng et al., 2021). GPR is a powerful state-of-the-art probabilistic non-parametric regression method (Calandra et al., 2016; Schulz et al., 2018), which has produced satisfactory results regarding the prediction of daily river temperature (Zhu et al., 2018; Grbić et al., 2013) but has rarely been used for air temperature estimation. In this study, we utilized the RF, SVM, and GPR machine learning methods to develop a model for interpolation of long-term air temperature data for China.

The ultimate objective of the study is production of a long-term high-resolution spatially continuous monthly air temperature product for China, based on meteorological station data and the best-performing model constructed using the machine learning techniques. The specific variables contained in the generated product include monthly mean air temperature (Tmean), monthly maximum air temperature (Tmax), and monthly minimum air temperature (Tmin) from January 1951 to December 2020 across China.

## 2 Data

### 2.1 Meteorological station data

Observational data of monthly Tmax, Tmin, and Tmean recorded from January 1951 to December 2020 at meteorological stations distributed across China were downloaded from the China Meteorological Data Service Centre (https://data.cma.cn/data/, last access: 17 April 2022). The height of the air temperatures from the weather stations is 2 m above the ground. The dataset includes information from 613 stations, which were split randomly into a training set (70%) for model training and a testing set (30%) for model evaluation (Figure 1). The data division was implemented by the "Subset Features" (Geostatistical Analyst) tool in ArcGIS referring to previous studies (Costache et al., 2020; Band et al., 2020; Mohajane et al., 2021; Kutlug Sahin and Colkesen, 2021). The number of weather stations in different years was not always exactly 613; the early years of the 1950s had notably fewer stations available (See Figure S1 for further details regarding the number of weather stations in each year and the data records each station contains). Note that we did not impute the missing data to make all the stations have all the monthly temperature data from Januray 1951 to December 2020. The source station meteorological data is quality-controlled and adjusted and the stations with no data are deleted in the study.

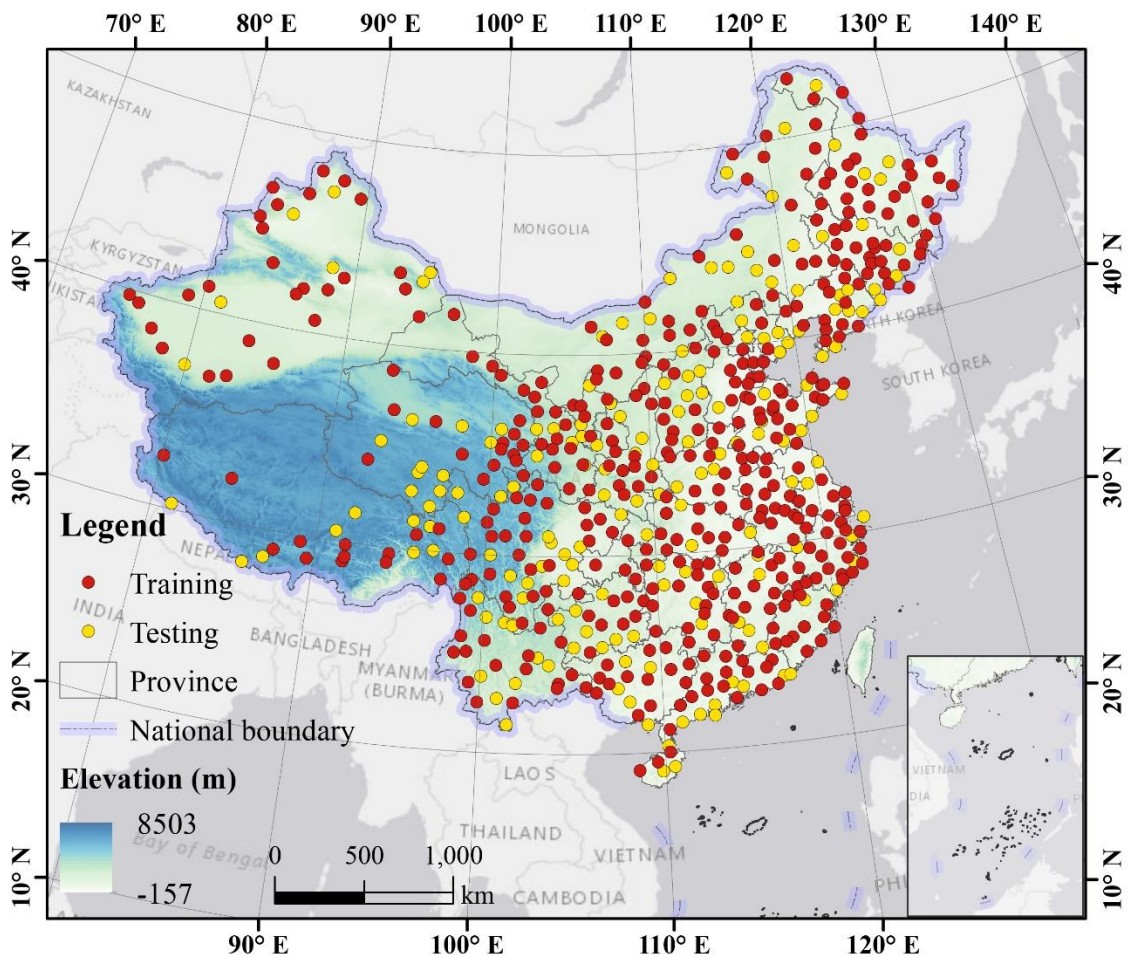

**Figure 1: Elevation and spatial distribution of meteorological stations across China (70% were used for training; 30% were used for testing).**

## 2.2 Topographic data

The topographic data used in this study comprised a digital elevation model (DEM) obtained from the NASA Shuttle Radar Topographic Mission (SRTM) (https://srtm.csi.cgiar.org/, last access: 15 July 2021). We used STRM version 4, which is the latest SRTM DEM product. The spatial resolution of the DEM is 3 arc seconds (approximately 90 m resolution). The DEM was resampled to 1 km resolution using the nearest neighbour method to produce the air temperature dataset with 1 km resolution. Gridded latitudinal and longitudinal coordinates of $1 \times 1$ km pixels were also used as components. All data used in this study were processed in the WGS84 Geographic Coordinate System (EPSG:4326).

## 2.3 Existing temperature products for comparison

We used three existing temperature products for comparison: 1) the Monthly Climate and Climatic Water Balance for Global Terrestrial Surfaces, University of Idaho (TerraClimate) dataset (resolution: 2.5 arc minutes, about 4.6 km) (https://developers.google.com/earth-engine/datasets/catalog/IDAHO_EPSCOR_TERRACLIMATE, last access: 15 July 2021); 2) the Famine Early Warning Systems Network (FEWS NET) Land Data Assimilation System (FLDAS) dataset (resolution: 0.1 arc degrees, about 11 km) (https://developers.google.com/earth-engine/datasets/catalog/NASA_FLDAS_NOAH01_C_GL_M_V001, last access: 15 July 2021); and 3) the latest climate reanalysis produced by the ECMWF/Copernicus Climate Change Service (ERA5 Monthly aggregates) dataset (resolution: 0.25 arc degrees, about 28 km) (https://developers.google.com/earth-engine/datasets/catalog/ECMWF_ERA5_MONTHLY, last access: 15 July 2021 ). The three datasets were used for comparison with our derived gridded temperature data. TerraClimate was used for comparing Tmax and Tmin using the maximum temperature (tmmx/°C) and the minimum temperature (tmmn/°C) variables, respectively. FLDAS was used for comparing Tmean using the near-surface air temperature variable (Tair_f_tavg/K), and we converted the unit (K) into degrees Celsius. ERA5 was used for comparing Tmax, Tmin, and Tmean using the average air temperature at 2 m height (mean_2m_air_temperature/K), maximum air temperature at 2 m height (maximum_2m_air_temperature/K), and minimum air temperature at 2 m height (minimum_2m_air_temperature/K), respectively, and the unit (K) was converted into degrees Celsius. The available time periods for the TerraClimate, FLDAS, and ERA5 products on the Google Earth Engine platform are 1958-01-01 to 2020-12-01, 1982-01-01 to 2021-05-01, and 1979-01-01 to 2020-06-01, respectively. Considering the overlapping periods, we chose January 1979 to December 2019 for the comparisons of Tmax and Tmin, and the period January 1982 to December 2019 for the comparisons of Tmean. The height of the temperature data from FLDAS is also 2 m (McNally et al., 2017). For TerraClimate data, it is produced based on other datasets including WorldClim, CRUTs4.0 and JRA-55 (Abatzoglou et al., 2018, p.1958–2015). The temperatures in WorldClim are at 2 m height (Fick and Hijmans, 2017; Chou et al., 2020). The temperature data from CRU Ts and JRA-55 are also at 2 m height (Harris et al., 2020). Therefore, the TerraClimate dataset also represents the 2m temperature.

## 3 Methods

### 3.1 Variable selection

The spatial distribution of air temperature is closely related to latitude, longitude, and elevation (Shao et al., 2012). The use of such auxiliary data can help alleviate to a certain extent the limitation of spatial interpolation associated with the sparse and irregular distribution of meteorological stations and increase estimation accuracy (Chen et al., 2015; Alvarez et al., 2014; Li and Heap, 2011; Newlands et al., 2011). Figure 2 displays the correlation coefficients between air temperature (i.e., Tmax, Tmin, and Tmean) and the above three geographical variables. Note that the correlation coefficient value for each month represents the average of all years (1951–2020), which was obtained based on all the observed data from meteorological

stations. The box plots of the correlation coefficient for each month are provided in Figure S2. Overall, Tmax, Tmin, and Tmean have positive (negative) correlations with respect to longitude (latitude and elevation). Longitude and elevation have opposite correlations but a similar trend with Tmax, Tmin, and Tmean, i.e., reasonably high correlation during summer (June–August) and low correlation during winter (December–February). Latitude is correlated negatively with Tmax, Tmin, and Tmean, i.e., strong (weak) correlation in winter (summer). It is evident that strong regularity exists in the relationships between air temperature and longitude, latitude, and elevation. In the subregions of the mainland China, the relationships between temperature and the variables still hold. Thus, we chose the three variables as predictor variables for obtaining the gridded temperature raster from the point observations. Owing to the incompleteness of remote sensing data attributable to imaging time constraints and cloud contamination,we did not consider satellite-derived independent variables. We considered only longitude, latitude, and elevation as predictor variables to give the derived model the advantages of ease of use, generalizability, and universality.

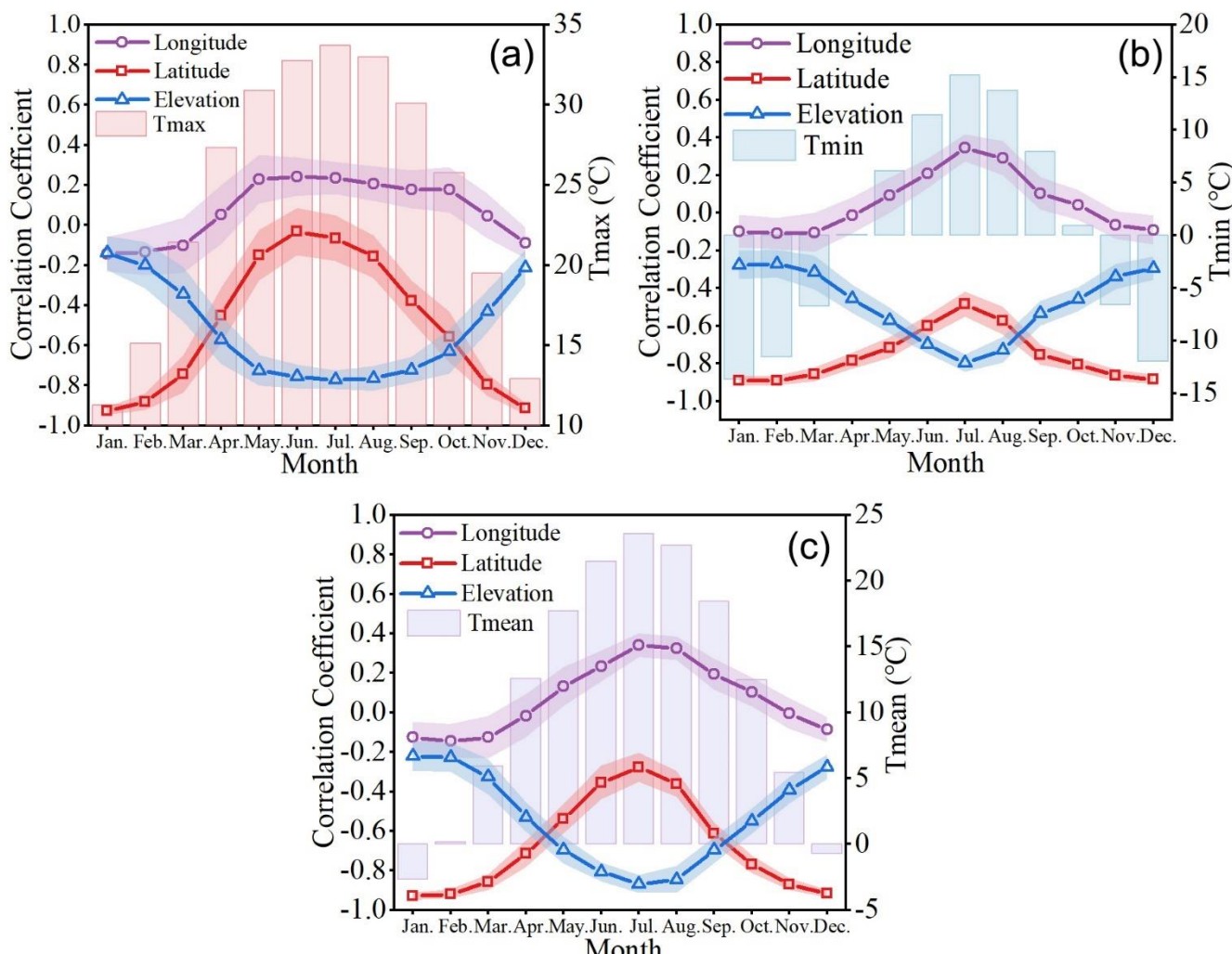

**Figure 2: Correlation coefficients between (a) monthly maximum air temperature (Tmax), (b) minimum air temperature (Tmin), and (c) mean air temperature (Tmean) and longitude, latitude, and elevation for each month. Coloured shading indicates the standard deviation. Note that the correlation coefficients are the average values of the correlation coefficients for each month over 70 years (1951–2020).**

## 3.2 Machine learning models

### 3.2.1 Random forest (RF)

RF, proposed by Breiman (2001), has been used widely for regression of geographical variables. RF is an ensemble machine learning method that consists of multiple decision trees. RF can produce high rates of accuracy, and the performance of RF in predicting new data is determined by the aggregation of the results of all the trees (Hengl et al., 2018). The randomization of RF lies in two aspects: the random selection of training samples for a tree through bagging (a form of bootstrapping), and the

random selection of predictor variables as the splitting attributes at each node of the tree (Merghadi et al., 2020; Yoo et al., 2018). The randomness of RF makes it resistant to the problem of overfitting. RF, which has been demonstrated promising and flexible in dealing with heterogeneity in the geographical environment, has been applied to prediction of spatial and temporal variables (Hengl et al., 2018; Zeng et al., 2021; Yoo et al., 2018). For further detailed information regarding RF, the reader is referred to Breiman (2001). We used the ensemble algorithm for regression in MATLAB R2020b for the RF implementation and used the default parameters. The minimum observations per leaf were set at 8 and the number of ensemble learning cycles was set at 30. The reader is referred to the MATLAB help centre for further details (https://www.mathworks.com/help/stats/fitrensemble.html?searchHighlight=NumLearningCycles&s_tid=srchtitle, last access: 15 July 2021; https://www.mathworks.com/help/stats/ensemble-algorithms.html, last access: 15 July 2021; https://www.mathworks.com/help/stats/fitrensemble.html#bvcj_t2-15, last access: 15 July 2021).

### 3.2.2 Support vector machine (SVM)

SVM, developed by Vapnik (2013), utilizes the inductive principle of structural risk minimization to obtain the overall optimal response. SVM transforms input data from lower-dimensional into a high-dimension space based on a series of kernel functions (Fan et al., 2018). The input space and the output space are non-linearly related in real applications, and the limitation is solved by mapping the input space on to higher dimension. In regression applications, an optimal hyperplane is constructed that is as close to as many samples as possible. The SVM does not only consider the error approximation to the data but also the model generalization. SVM has been used widely in various fields such as meteorology, hydrology, and agriculture for regression and prediction applications (Ghorbani et al., 2017; Shrestha and Shukla, 2015; Fan et al., 2018). Detailed information regarding SVM can be found in Vapnik (2013). The Gaussian kernel was adopted as the kernel function of SVM and the kernel scale parameter was set to 1.7. The value of Epsilon is an estimate of a tenth of the standard deviation using the interquartile range of the response variable (by default). The box constraint value for the Gaussian kernel function was obtained by dividing the interquartile range of the response variable by 1.349. The box constraint and the epsilon hyperparameters are varying from month to month according to the training data of each month. The predictors were standardized in the SVM model. The reader is referred to the MATLAB help documentation for further technical details (https://www.mathworks.com/help/stats/fitrsvm.html, last access: 15 July 2021 and https://www.mathworks.com/help/stats/understanding-support-vector-machine-regression.html, last access: 15 July 2021).

### 3.2.3 Gaussian process regression (GPR)

GPR is a non-parametric Bayesian technique for solving nonlinear regression problems (Grbić et al., 2013). GPR was originally proposed to provide a "principle, practical, and probabilistic approach to learning in kernel machines" (Rasmussen, 1997, 2004). GPR is based on Bayesian theory and statistical learning theory, which is applicable to regression problems (Zhang et al., 2019). GPR has strength in its seamless combination of several machine learning tasks such as model training, hyperparameter estimation, and uncertainty estimation, which can compute the prediction intervals using the trained model

(Sun et al., 2014; Zhu et al., 2018). GPR has been utilized in diverse applications that include model approximation, experiment design, and multivariate regression (Zhu et al., 2018; Karbasi, 2018); however, previous application of GPR to prediction of air temperature has been limited. For detailed information regarding the GPR model, the reader is referred to Rasmussen (1997, 2004). The explicit basis in the GPR model is "constant" and the kernel function of the GPR algorithm is the exponential kernel. The predictor variables were standardized in the GPR model. The reader is referred to the MATLAB help documentation for further details regarding GPR (https://www.mathworks.com/help/stats/fitrgp.html, last access: 15 July 2021 and https://www.mathworks.com/help/stats/gaussian-process-regression-models.html, last access: 15 July 2021).

We extracted the independent variables (i.e., latitude, longitude, and elevation) relating to the meteorological stations and randomly divided the processed data into a set for model training (70%) and a set for model evaluation and validation (30%). For each month, we used the temperature data of a month (training set) to train the model and then used this model to generate the grid data of the same month. When training the models, the 10-fold cross-validation was used. We constructed a model for each month separately which means we have 840 models for the 840 months from 1951 to 2020.

### 3.3 Model evaluation metrics

We used three metrics to evaluate model performance: mean absolute error (MAE), root mean square error (RMSE), and the coefficient of determination ($R^2$), which have all been used widely in previous studies to evaluate model capability in predicting the dependent variable (Graf et al., 2019; Khanal et al., 2018; Peng et al., 2019; Ji et al., 2015). The MAE is the mean value of all the individual errors. The RMSE measures the discrepancy between the observed and predicted values. The MAE and RMSE both summarize the mean difference between the observed and predicted values and are among the best overall measures of model performance (Li and Heap, 2011). Lower values of MAE and RMSE mean better accuracy. $R^2$ measures the proportion of variance explained by the model (Sekulić et al., 2021), representing how well the predicted values fit in comparison with the observed values. The higher the $R^2$ value, the better the model performance:

$$\text{MAE} = \frac{1}{n}\sum_{i=1}^{n} |P_i - O_i|\,, \tag{1}$$

$$\text{RMSE} = \sqrt{\frac{1}{n}\sum_{i=1}^{n} (P_i - O_i)^2}\,, \tag{2}$$

$$R^2 = 1 - \frac{\sum_{i=1}^{n} (O_i - P_i)^2}{\sum_{i=1}^{n} (O_i - \bar{O})^2}\,, \tag{3}$$

where $P_i$ is the predicted value in the time series, $O_i$ refers to the observed value from the meteorological stations, $n$ is the number of samples, and $\bar{O}$ represents the average of the observed values from $n$ meteorological stations. All performance measures were calculated using the testing dataset for evaluation purposes.

### 3.4 Methods for spatiotemporal analysis of monthly air temperature

The Theil–Sen slope estimator used in combination with Mann–Kendall (MK) detection, which is an effective approach for trend analysis that reflects the variation in trends of each pixel in a time series, has been used widely in various fields such as hydrology and meteorology (Cai and Yu, 2009; Gocic and Trajkovic, 2013; Jiang et al., 2015). In this study, we used the Theil–Sen estimator coupled with the MK test to detect the trend of the temperature time series.

**(1) Theil–Sen estimator**

The Theil-Sen estimator, which is a robust non-parametric approach for estimating the slope of a trend, has been used widely in relation to hydrometeorological time series data (Jiang et al., 2015; Gocic and Trajkovic, 2013; Shifteh Some'e et al., 2012; Sayemuzzaman and Jha, 2014). The Theil–Sen slope estimator, which represents the magnitude of a trend, can be expressed as: (Theil, 1950; Sen, 1968):

$$\beta = \text{Median}\left(\frac{x_j - x_i}{j - i}\right), \forall j > i, \tag{4}$$

where $\beta$ denotes the Theil–Sen median slope, and $x_i$ and $x_j$ refer to the air temperature at time $i$ and $j$, respectively. The slope derived from the Theil–Sen estimator is a robust estimate of the magnitude of a trend, which can represent an increasing trend ($\beta > 0$) or a decreasing trend ($\beta < 0$) over the study period on the pixel scale. In this study, the Theil–Sen median slope was computed using the MATLAB platform.

**(2) Mann–Kendall (MK) test**

The MK test quantifies the significance of a trend. It is a non-parametric statistical test, meaning that it does not require samples to follow specific distributions and is not influenced by outliers. The MK test has frequently been applied to measure the significance of trends in hydrological and meteorological time series data (Jiang et al., 2015; Shifteh Some'e et al., 2012; Da Silva et al., 2015; Gocic and Trajkovic, 2013). The Z statistic is used to evaluate a trend; a positive (negative) value of Z means an increasing (decreasing) trend. Further details regarding the MK test can be found in Jiang et al. (2015) and Shifteh Some'e et al. (2012). In this study, we set the significance level at 5%, the same as many other related studies (Jiang et al., 2015; Shifteh Some'e et al., 2012; Da Silva et al., 2015), which means the variation is significant when |Z| is >1.96; otherwise, the variation is non-significant. The MK test was conducted using MATLAB language.

## 4 Results

### 4.1 Evaluation of model performance

We used the testing dataset to evaluate the performance of each model. Figure 3(a)–(c) presents the MAE, RMSE, and $R^2$ values of Tmean, respectively, of the three machine learning models for each month in the time series of 1951–2020. The MAEs of GPR and SVM are close to 1°C across the study period (the MAEs are slightly smaller for GPR), while the MAEs of RF are clearly higher than those of both GPR and SVM. The RMSEs have the same order as the MAEs, i.e., GPR outperforms both SVM and RF. The differences in the RMSEs of the three models are evident; GPR has the lowest RMSE in

every month throughout the study period (maximum RMSE = 1.35°C, average RMSE = 0.79°C, and Std = 0.15°C). Detailed inspection of the MAEs and RMSEs from January 2015 to December 2020 (Figure S3 in the Supplementary Material) reveals that the errors are relatively larger in cold months (November–February) and smaller in warmer months. All three models show relatively high values of $R^2$. GPR and SVM have $R^2$ values that are very similar, i.e., average $R^2$ values of 0.97 and 0.96, respectively, while RF has lower values of $R^2$, especially during the first few years. For Tmean, RF shows distinct fluctuations throughout January 1951 to December 2020, whereas GPR and SVM are relatively stable. The accuracy metrics show that the MAEs and RMSEs fluctuate from month to month, while $R^2$ remains reasonably constant. The accuracy metrics of GPR averaged over 840 months from January 1951 to December 2020 are as follows: MAE = 0.79°C, RMSE = 0.79°C, and $R^2$ = 0.97 for Tmean. The three metrics indicate that GPR always has the highest accuracy and lowest standard deviation, reflecting the robustness of GPR. For Tmax and Tmin, GPR still performs best according to the evaluation metrics (Figs. S4 and S5). The correlation coefficients of air temperature and the predictor variables (Figure 2) vary from month to month, which might contribute to the fluctuation in the accuracy of the interpolation with month.

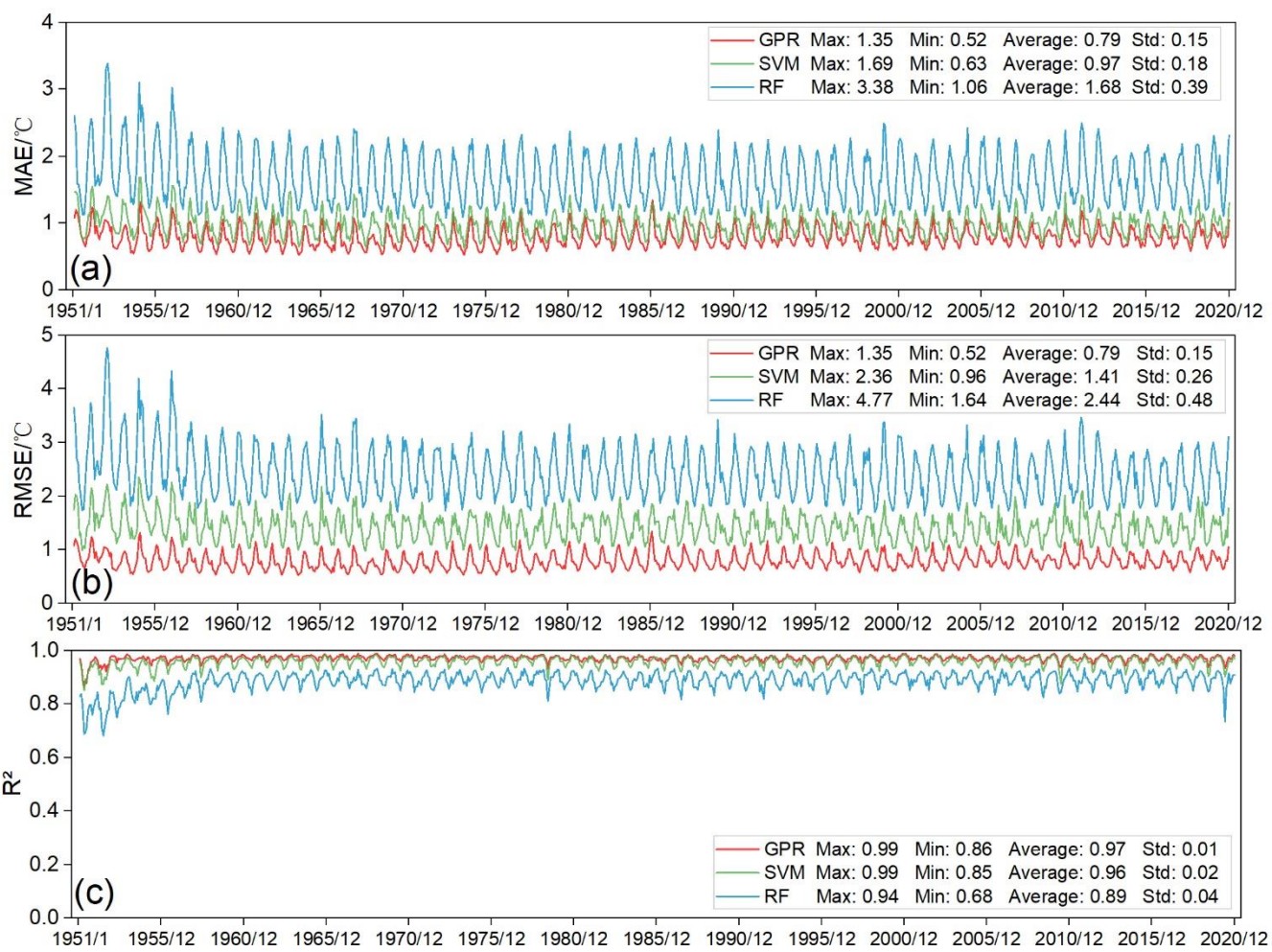

**Figure 3: (a) Mean absolute error (MAE), (b) root mean square error (RMSE), and (c) coefficient of determination ($R^2$) between observed Tmean and predicted Tmean by the three machine learning models (GPR, SVM, RF) of the test meteorological stations over the period from January 1951 to December 2020. See Figure S4 and Figure S5 for the accuracy graph of Tmax and Tmin.**

The residuals were obtained as the observed values minus the predicted values. Figure 4 shows box plots of the residuals

for Tmean for the test meteorological stations each month during 1951–2020. Overall, the mean residuals of the three models are generally close to 0, and the residuals are smaller during the warm months (June–September) than during the cool/cold months (October–April), particularly for RF and SVM. In comparison with SVM and RF, GPR has the most stable accuracy over the 12 months, i.e., the difference in the residuals among the months is relatively small. GPR also has a quantile range that is narrower than that of the other models. For Tmax and Tmin, the bias of GPR over the 12 months is smaller than that of

both RF and SVM (Figures S6 and S7). Additionally, the accuracy of the estimated Tmax is higher than that of Tmin, consistent with the findings of Tang et al. (2020a). The results show that the GPR model could be a better choice than either RF or SVM for estimating Tmean, Tmax, and Tmin for China. The frequency distributions of the residuals of the three machine learning

models for Tmean, Tmax, and Tmin are provided in the Supplementary Material (Figures S8–S10), in which it can be found that GPR generally has the greatest concentration of residuals close to 0.

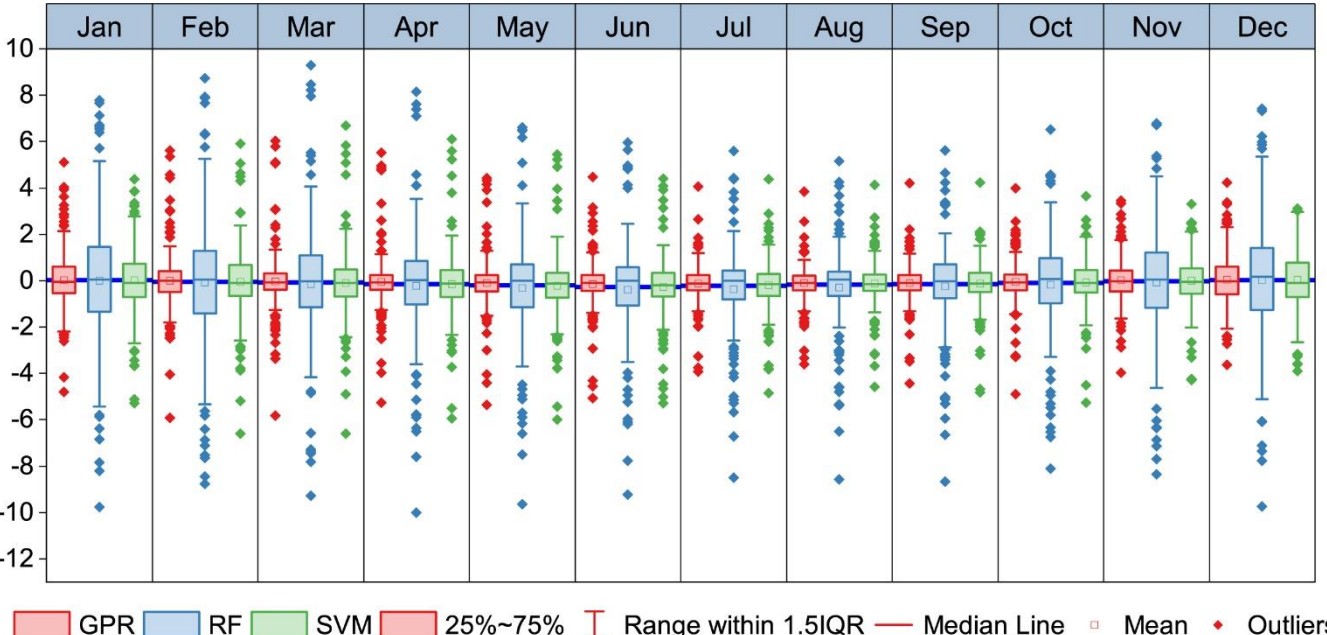

**Figure 4: Residuals of the monthly Tmean predicted by the machine learning models with respect to in situ Tmean for the test meteorological stations. Note that the average of the residuals of Tmean from 1951–2020 for each test meteorological station is shown for each month. See Figure S6 and Figure S7 for the residual graph of Tmax and Tmin.**

The spatial distribution of the average values of the residuals of the GPR results for Tmean throughout the 70 years (1951–2020) at each of the test meteorological stations is displayed in Figure 5. Most areas have relatively low absolute residuals, although certain stations in some western areas have relatively high residuals. In January and December, the number of stations with high absolute residuals (>2.5°C) is relatively higher than that in other months, i.e., 13 and 12 stations, respectively. Conversely, there are only five, five, and four stations with absolute residuals >2.5°C for June, July, and September, respectively. This might indicate that the GPR model produces better results during warmer months. Furthermore, among the stations with high absolute residuals (>2.5°C), more are positive than negative, indicating that the observed values are higher than the predicted values, i.e., there is slight underestimation by GPR at those stations. Overall, most stations show residuals between −1°C and 1°C. The maps of the residuals for Tmax and Tmin also display patterns that are spatially similar to the maps of residuals for Tmean; however, the overall residuals of Tmax exhibit better results in comparison with the spatial pattern of the residuals of Tmin (Figures S11 and S12).

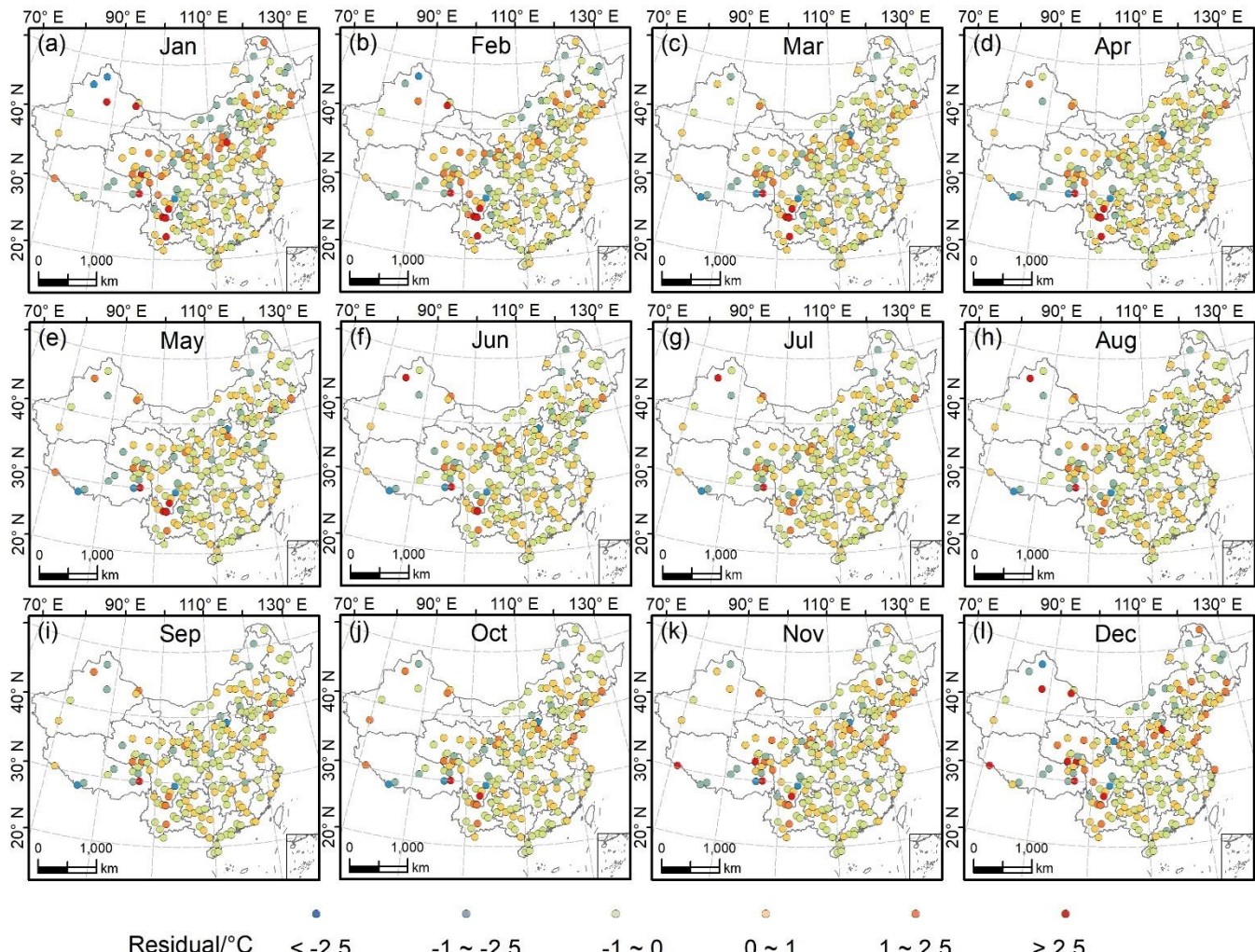

**Figure 5: Spatial distribution of residuals between the observed Tmean and the predicted Tmean by GPR for the test meteorological stations for each month. Note that the exhibited residuals are the average residual of 70 years (1951–2020) for each month.**

## 4.2 Spatial distribution of air temperature

According to the model evaluation, we concluded that GPR is the best model for estimating air temperature across China. Therefore, we employed the GPR model to generate the long-term spatial dataset of Tmean, Tmax, and Tmin from January 1951 to December 2020, which we named GPRChinaTemp1km. Figure 6 illustrates the spatial pattern of Tmean estimated by GPR in 2020. The differences between northwestern and southeastern regions are remarkable. Generally, Tmean decreases from the southeast toward the northwest. In winter, the temperature range between northern and southern China is large, whereas the temperature range in summer is relatively small. The lowest Tmean (−27°C) occurs in January and the highest Tmean (34°C) occurs in July, consistent with the fact that January and July are generally the coldest and hottest months, respectively. The maps show reasonable changes as the seasons change, i.e., high temperatures in summer (June–August) and

320 low temperatures during winter (December–February). Overall, Tmax and Tmin in China follow a pattern similar to that of Tmean, i.e., decreasing from the south toward the north (Figs. S13 and S14). The highest Tmax of 2020 (44°C) occurs in July (Figure S13) and the lowest Tmin (−43°C) occurs in December (Figure S14). The results well describe the spatial heterogeneity of air temperature across China. Additionally, the border of the Tibetan Plateau is evident in the maps of Tmax, Tmin, and Tmean for each month, especially in the winter and summer seasons, further demonstrating the rationality of the derived results.

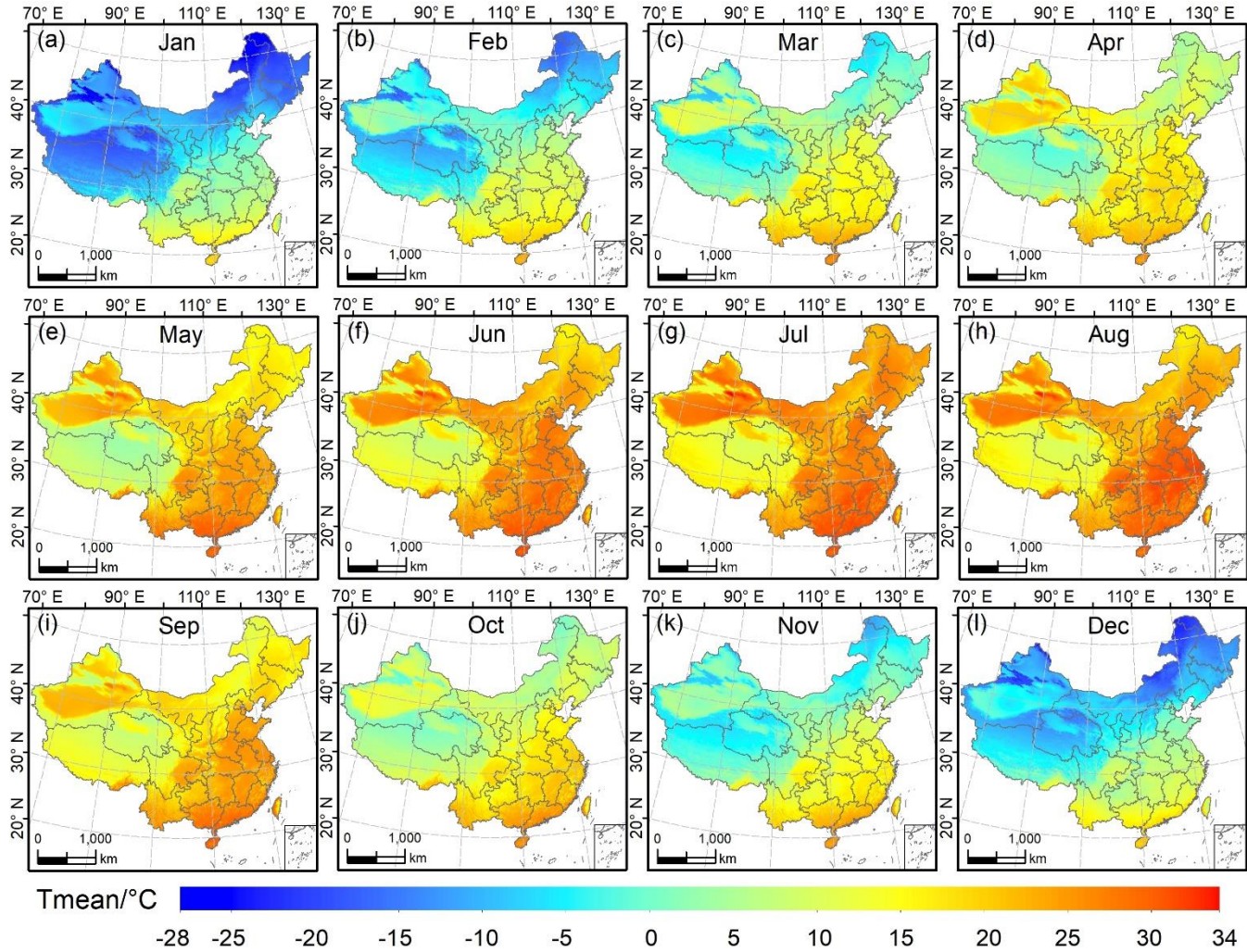

**Figure 6: Spatial distribution of monthly Tmean predicted by GPR across China for each month in 2020. Note that only the maps for 2020 are presented as an example (all the data are available in the China GPRChinaTemp1km database).**

### 4.3 Trend analysis of air temperature in China

Theil–Sen median trend analysis was integrated with the MK test and the results were classified into four categories: significant
increase, non-significant increase, significant decrease, and non-significant decrease. Figure 7 shows that the trend of the variation of Tmean (1951–2020) in China is dominated by significant increase in each month. There is only a small region in

northwestern China that has significant decrease in Tmean in January and December. We found that there is always a small region showing a different trend in comparison with surrounding areas in the Xinjiang Uygur Autonomous Region in northwestern China, which is characterized by a decreasing trend in most months and non-significant increase in the hot months (June–September). This phenomenon could be related to the complex conditions of the region. For example, Bayinbuluke is an intermountain basin surrounded by the Tianshan Mountains with an alpine wetland ecosystem in the arid temperate zone. During summer (June–August), Tmean shows distinct non-significant decrease in central areas of China. In December, the spatial differentiation is the most remarkable, and the increasing trend in most of eastern China is non-significant, which differs from that of other months, and there is a region representing a trend of non-significant decrease on the Yungui Plateau in southwestern China. Overall, the trend of Tmean in China during 1951–2020 shows significant increase in each month, while only a few areas have a trend of decrease. The distribution of the mean temperature trend in China in our study agrees with the existing literature (Dong et al., 2015, p.1963–2012; Sun et al., 2018; You et al., 2021; Cui et al., 2017, p.1960–2015). Tmax is characterized by significant increase and non-significant increase, as well as a non-significant decreasing trend (Figure S21). Tmin exhibits a spatial pattern similar to that of Tmean, showing a significant increasing trend in most areas in each month (Figure S22).

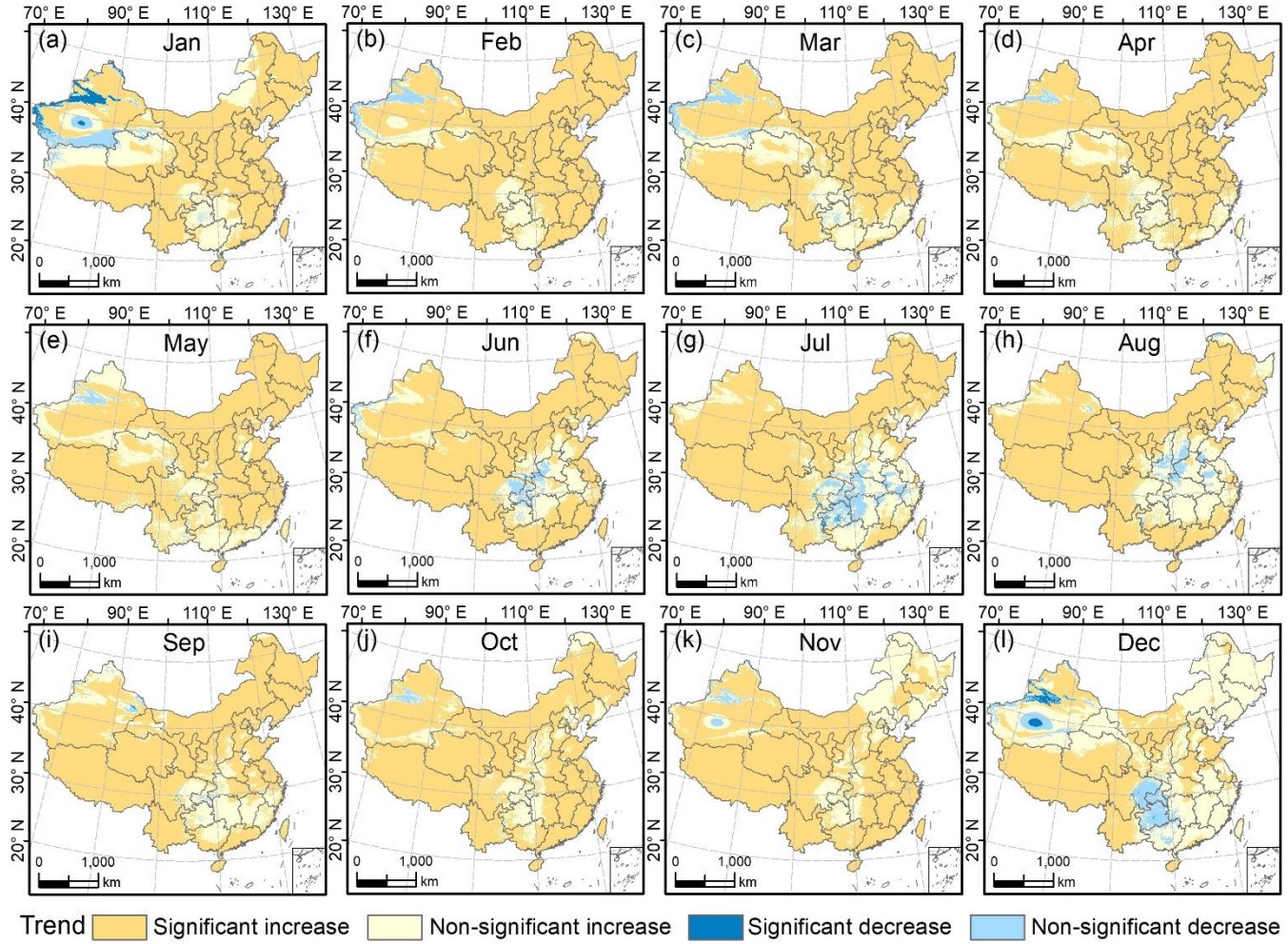

**Figure 7: Monthly trends of Tmean change in China during 1951–2020 obtained by Theil–Sen median slope analysis. The significance of the trends is quantified by the Mann–Kendall statistical test at the 95% confidence level. The separate Theil–Sen trend analysis and MK test results for Tmean, Tmax, and Tmin are provided in the Supplementary Material (Figures. S15–S20).**

## 5 Discussion

### 5.1 Comparison with traditional interpolation methods

Two traditional methods used widely for spatial interpolation are IDW and OK (Li and Heap, 2014, 2011). In this study, we used ANUSPLIN in addition to IDW and OK for comparison with the machine learning models. ANUSPLIN, which is professional interpolation software that uses the thin-plate smoothing spline algorithm (Hutchinson, 1995, 2004; Xu and Hutchinson, 2013), has been used to create many climatic datasets such as the monthly Climatic Research Unit dataset (New et al., 2000) and the WorldClim dataset (Fick and Hijmans, 2017; Hijmans et al., 2005). We compared the interpolation results derived using the machine learning models with the results obtained using the traditional methods to further assess the

interpolation power of the machine learning methods regarding air temperature across China. The accuracy metrics (Figure 8)

show that the performances of GPR, SVM, and ANUSPLIN are of a similar level, while RF, IDW, and OK perform less well. Both IDW and OK have relatively high interpolation errors with higher MAEs and RMSEs than GPR and SVM (Figure 8). Overall, IDW and OK do not perform well in July and January of all the studied years. Figure 9 shows scatter plots of observed monthly Tmean and Tmean estimated by the six models for January and July in 2020. It can be seen that OK and IDW both have clear differences between January and July (Figure 9g, h, j, and l), in which the points are relatively widely dispersed in

July. GPR, SVM, and ANUSPLIN are slightly affected by the seasonal variation with lower errors (i.e., lower MAEs and RMSEs) in July (Figure 9). As shown in Figure 3, the RMSE, MAE and $R^2$ show a cyclic pattern. In winter (Nov, Dec, Jan and Feb), temperature has relatively lower correlations with two variables (i.e., longitude and elevation), while in summer, temperature has lower correlation with only one variable (i.e., latitude) and has high correlations with longitude and elevation. The elevation can add the topographic information that can increase the temperature interpolation reliability (Rolland, 2003),

which may be a reason for the larger errors in the cold months (Amini et al., 2019; Brunetti et al., 2014; Stahl et al., 2006). The RMSE and MAE are high for the winter months as shown in the zoomed-in accuracy in Figure S3. This accuracy cycle pattern is probably induced by the correlation difference between summer and winter. GPR has the lowest MAEs and RMSEs, and the highest $R^2$ values in most months. Note that the RMSE and MAE values of ANUSPLIN for July months in 1970, 1980 1990, 2010 and 2020 are slightly lower than GPR (Figure 8). Considering the proven power of ANUSPLIN in predicting

meteorological variables, the GPR yields relatively satisfactory results. Taking the accuracy in 2020 as an example (Figure 9), ANUSPLIN has higher errors and lower $R^2$ values than GPR, and there are certain points with values estimated by ANUSPLIN that are relatively far away from the observed values in July (Figure 9l). In contrast, the Tmean values estimated by GPR are relatively close to those of the in situ Tmean values (Figure 9b).

Comparison of the performances of the six models for Tmax and Tmin reveals that GPR performs better in terms of Tmax

and has the lowest errors (MAEs and RMSEs) in almost all the studied months (Figure S23). OK and IDW have similar performances, consistent with the findings of previous related studies (Plouffe et al., 2015; Li et al., 2011a). It is noticeable that IDW and OK perform relatively poorly. Both IDW and OK depend on the spatial autocorrelation of air temperature and cannot capture the geomorphic characteristics of the interpolation area because neither method includes elevation information (Ozelkan et al., 2015; Wang et al., 2017; Li et al., 2011a). Unlike IDW and OK, ANUSPLIN considers longitude, latitude, and

elevation (Hijmans et al., 2005). The frequency distributions of the residuals for Tmean, Tmax, and Tmin of the six models for the same months as in Figure 9 are presented in the Supplementary Material (Figures S24–S26). The distributions follow a normal distribution, and the residuals of GPR, SVM, and ANUSPLIN are concentrated mainly around 0. Scatter plots of Tmean, Tmax, and Tmin for the same periods as shown in Figure 9 are provided in the Supplementary Material (Figures S27– S49), in which the robustness of GPR is clearly demonstrated for Tmean, Tmax, and Tmin in comparison with other methods.

Studies have shown that Gaussian processes are one of the most intuitive techniques for modelling spatial surfaces (Yu et al., 2017; Berger et al., 2001). Besides, we conducted an experiment by cutting out a region using a 300 km square; we used all

stations inside this square for testing and stations outside it for training models to estimate the robustness of the GPR models; the results indicated high accuracy and good robustness of the GPR models (Figures S50-S52).

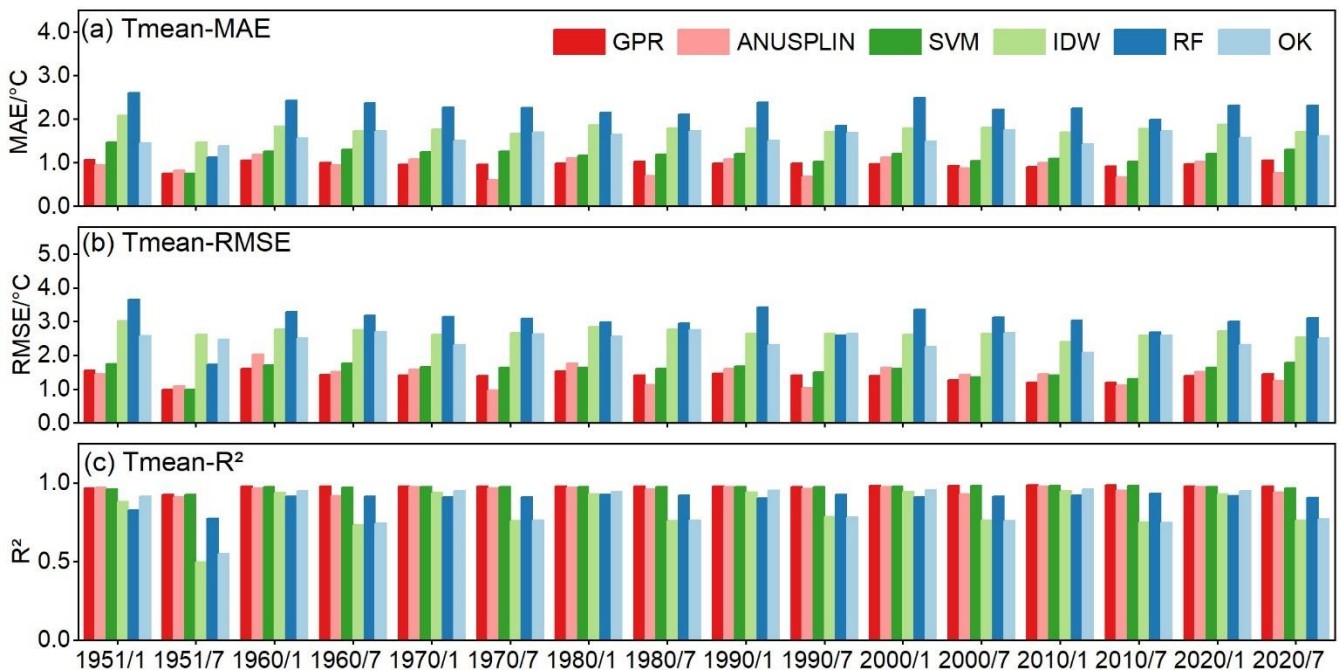

Figure 8: Accuracy of Tmean derived from the machine learning methods and traditional methods for January and July during 1951–2020 with an interval of 10 years.

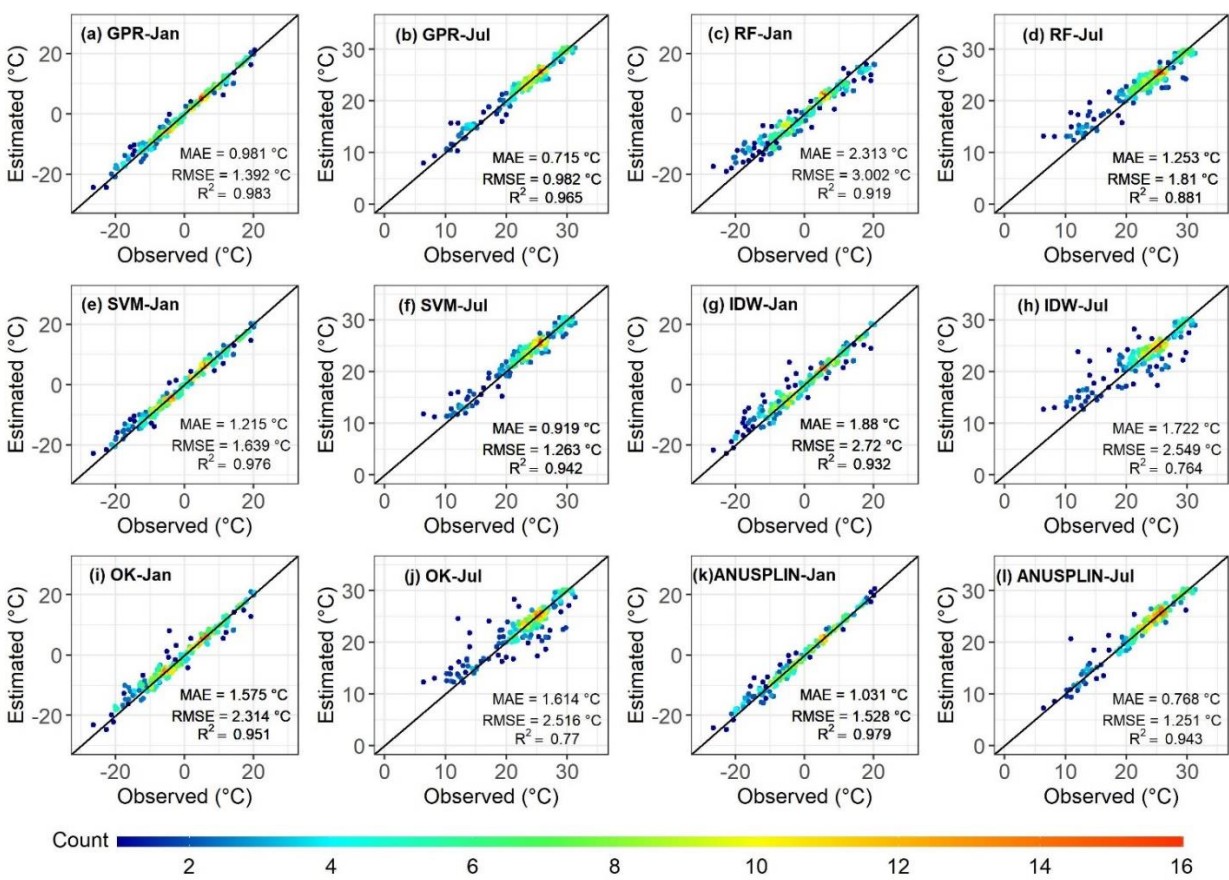

**Figure 9: Scatter plots of Tmean estimated by the machine learning models and traditional models against observed monthly mean temperature in January and July 2020.**

Figure 10 presents maps of the residuals (observed values minus estimated values) for Tmean in January and July 2020 estimated using the six methods. Bias is apparent in RF (Figure 10c and d), IDW (Figure 10g and h), and OK (Figure 10i and j). Comparison of the maps of the residuals reveals that Tmean estimated by GPR generally agrees well with the in situ data, with large bias at only a few stations that are distributed mainly in western and northern China, which might be related to the scarcity of meteorological stations and the complex regional topography (Ji et al., 2015). It is also evident that the absolute

residuals in July are generally lower than those in January (Figure 10). For China, the spatial homogeneity of temperature in summer is stronger than that in winter, which might be one reason for the lower bias observed in July. We note that RF has poor performance in comparison with the other machine learning methods. Although we do not have sufficient evidence to deduce the causes for the lower accuracy of RF, the small number of meteorological stations might be a major reason. Additionally, RF regression has a limitation regarding the conditions beyond the range of the training dataset because only the

values included in the training data are used for splitting the trees (Mutanga et al., 2012; Jeong et al., 2016). Among the three machine learning algorithms, GPR and SVM both perform relatively well, although the performance of GPR is better. Note

that we used the medium Gaussian SVM and exponential GPR in MATLAB R2020b. GPR and SVM are both non-parametric kernel-based models that rely on the Gaussian principle. The Gaussian function has the desired characteristics of being an inverse-distance algorithm and a smoothing filter (Thornton et al., 1997), which might explain the better performances of GPR

and SVM. The comparison with Peng's data (Peng et al., 2019) in the Tibetan plateau region shows the strength of the GPR data in regions with complicated topography (Figure S55).

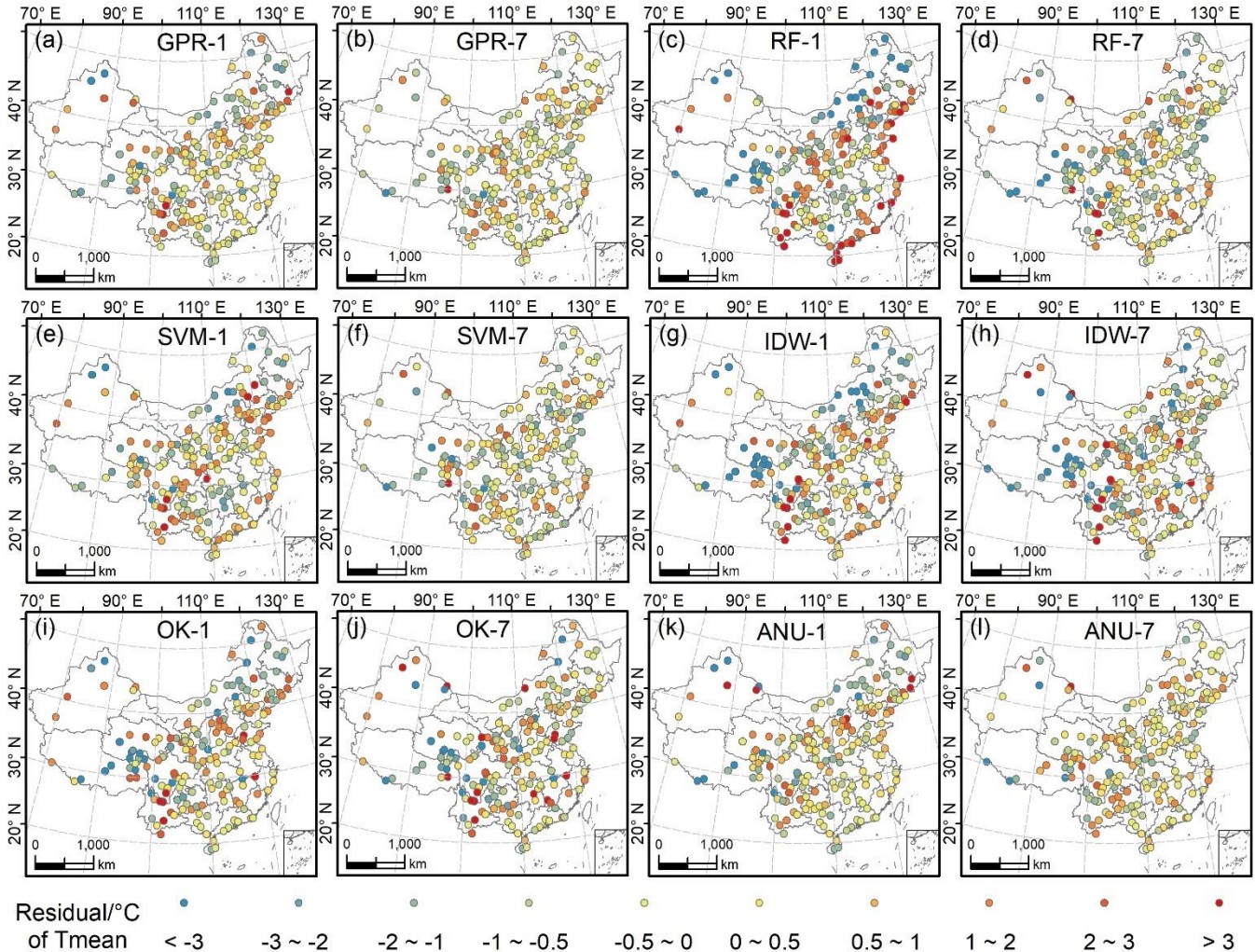

Figure 10: Comparison of the spatial distribution of the residuals between the machine learning methods and the traditional methods for Tmean in January and July 2020 (similar figures for Tmax and Tmin are provided in Figures S53 and S54).

In summary, ANUSPLIN is an interpolation method that is better than IDW and OK in modelling air temperature over complex terrain (Plouffe et al., 2015; Newlands et al., 2011); however, the robustness of ANUSPLIN is no better than that of GPR. Moreover, ANUSPLIN is based on the principle of thin-plate splines, the skill of which can be limited in regions with high elevations and sparse observations, i.e., areas such as the Tibetan Plateau (Jobst et al., 2017). Furthermore, in our study,

running ANUSPLIN was more time-consuming in comparison with running the GPR model, making it difficult to generate long-term monthly datasets for all 12 months over 70 years. The spatial maps of temperature generated by the six models (Figures S56–S58) reveal that GPR obtained reasonable results for Tmean, Tmax, and Tmin. In the case of Tmax and Tmin, ANUSPLIN does not appear to have a rational range for Tmin (Figure S58k and l). Therefore, in production of long-term high-resolution datasets over large land areas such as China, it is more feasible, efficient, and accurate to use the GPR model. Furthermore, the GPR generated data can capture the high temperature of the anomalous event (Figure S59), e.g., the 2006 summer drought of the eastern Sichuan Basin (Li et al., 2011c). The spatial anomaly pattern can also be captured using our generated gridded data, as shown in Figures S60-70. In our study, the GPR method is employed for generating the temperature data. In future work, we may dig into the potential of GPR in other meteorological variables.

## 5.2 Comparison with other products

We used the ERA5, FLDAS, and TerraClimate temperature datasets for comparison with our dataset generated using the GPR model. The spatial resolution of the three datasets is about 28 km, 11 km, and 4.6 km, respectively. Our datasets are 1-km. We resampled all the data to the resolution of ERA5 dataset to keep the resolution consistent and then we made comparisons. Note that the generated data in our study represent the temperature at 2 m height since the station records the temperature at 2 m above ground (Liu et al., 2011; Zhang et al., 2010). Taylor diagrams were constructed to compare the accuracy between our data and that of the other products for Tmax, Tmin, and Tmean (Figure 11). For Tmax, it can be seen that the GPR-simulated air temperature best matches the observations, with a closer standard deviation to the observed variability, lower centred RMSE, and higher correlation than both ERA5 and TerraClimate. For Tmin, the standard deviation and RMSE values of ERA5 are clearly greater than those of both TerraClimate and GPR. GPR has the almost same standard deviation as the observations with the lowest RMSE and highest correlation, whereas TerraClimate has slightly less spatial variability (lower standard deviation) with a higher RMSE value and lower correlation. In the case of Tmean, GPR and FLDAS have almost the same variability (with a standard deviation close to the observed variability), while GPR has the highest correlation and lowest RMSE. Generally, the GPR-derived dataset is better in terms of Tmax, Tmin, and Tmean than the datasets obtained using other products. The better outcome using the GPR model is characterized by the closest distance in terms of the variability compared with the observations, the lowest RMSE, and the highest correlation for all three temperature variables. The Taylor diagrams also show that the GPR model performs better in terms of the reliability of the gridded temperature datasets and has greater potential regarding spatial interpolation of air temperature. Besides, we also compared our datasets with Peng's data (Peng et al., 2019), which shows the mean temperature from GPR datasets has relatively higher accuracy than that from Peng's data on the whole, especially in warm months (Figure S71). Additionally, the high-resolution GPR data can provide more spatial details than the coarse resolution products like ERA and FLDAS (Figure S72).

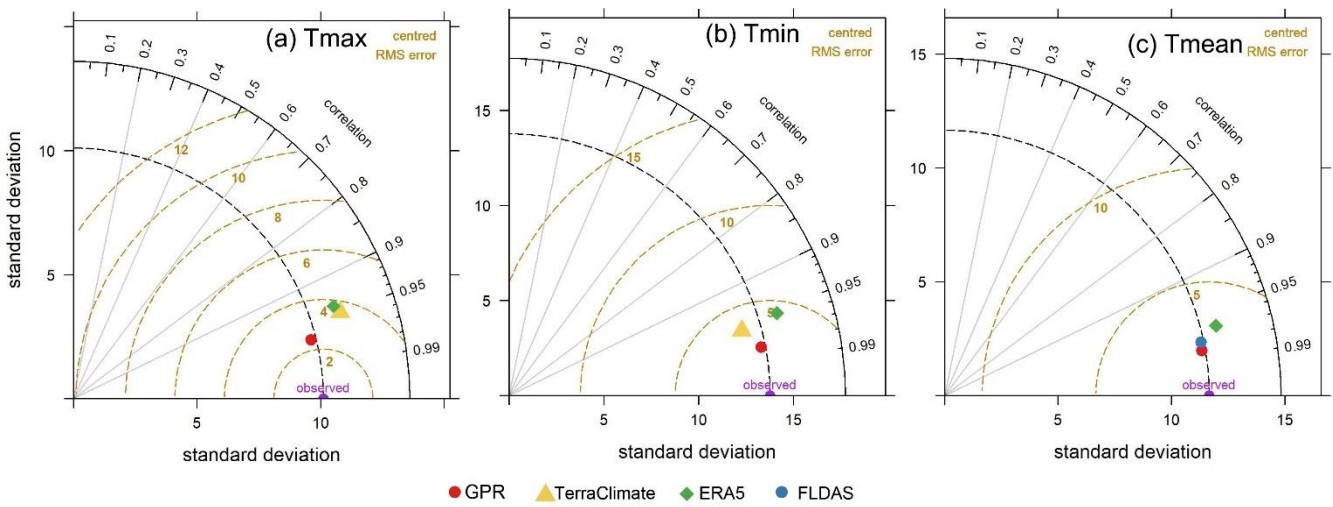


**Figure 11: Taylor diagrams displaying a statistical comparison with observations between our products generated using the GPR model and the other products under the same spatial resolution. Given the overlapping time of the datasets, January 1979 to December 2019 was used for comparing Tmax and Tmin and January 1982 to December 2019 was used for comparing Tmean. Comparisons for each month are presented in the Supplementary Material (Figures S73–S75).**

**5.3 Limitations**

China covers a vast territory with complex topography and diverse climate, meaning that auxiliary data such as elevation are particularly important regarding temperature interpolation (Appelhans et al., 2015; Vicente-Serrano et al., 2003). Air temperature is strongly impacted by topography, and the DEM represents a fundamental variable for interpolating air temperature in our methodology. The terrain semantics can be learned from the elevation data (Sha et al., 2020). The quality

of auxiliary environmental predictors is vital and an appropriate DEM is crucial for accurate interpolation (Li and Heap, 2011; Diodato, 2005). The DEM data adopted in this study were from the SRTM Version 4, which represents a substantial improvement on previous versions. Although the updated version is promoted as the highest quality SRTM dataset available (https://srtm.csi.cgiar.org/, last access: 15 July 2021), certain limitations remain. For example, Mukul et al. (2017) reported that the accuracy of the SRTM product in the region of the Himalayas decreases as elevation rises. Additionally, only a limited

number of external studies validating the SRTM version 4 product have been reported (Tan et al., 2015), and the uncertainty of the data in our application to air temperature interpolation should be assessed in future work.

The Euclidean distance of observation stations is quite small in most regions while it is relatively large in the west of the Tibetan Plateau and a small region in Inner Mongolia (Figure S76). The larger Euclidean distance means the stations in that region are sparse, which can have an impact on the interpolation accuracy (Hijmans et al., 2005; Li et al., 2011a). The spatial

distribution of the width of the predicted intervals with a significance level of 5% (using the upper limit minus the lower limit of the confidence interval) for the 12 months in 2010 using the trained models shows that most of the regions have quite small uncertainty while the Tibetan Plateau areas have relatively larger uncertainty (Figure S77).

Some studies use the remote sensing data to generate the air temperature dataset, such as land surface temperature, normalized differential vegetation index (NDVI), land-use (Hooker et al., 2018; Li and Zha, 2019; Li et al., 2018). Although these variables are correlated with the air temperature, these remote sensing data are usually not available before 2000 since our goal is to generate long term data series from 1951 to 2020. Furthermore, the MODIS data are not available for each month from January 2000 to December 2020. As shown in Figure S78, the percentage of the available MODIS images is low in northeast China and southern regions. Thus, the remote sensing data are not appropriate for generating long-term temperature data in our study. Furthermore, there is inherent data uncertainty in the remote sensing data itself, such as the land use data.

In our study, we split the stations into testing and training stations in ArcGIS, which has considered the spatial distribution of the weather stations. We also conducted a case study using the Tmean from 1990, 2000 and 2010 to figure out if the model output is sensitive to the choice of stations used in the test/training dataset. We conducted the experiment by randomly splitting the data into training and testing sets (7:3) 50 times in ArcGIS. The RMSE varies slightly from different scenarios of the test/training dataset, while there is no obvious variation in $R^2$ (Figures S79 and S80).

It should be noted that in July 1951, there were only 38 samples available for testing and 96 samples available for training. The scarcity of meteorological stations in the early years of the 1950s represents one of the major limitations regarding the use of the machine learning methods. Generally, this study found that GPR estimates Tmean better than Tmax and Tmin. The average MAEs and RMSEs of the GPR model for Tmean are both 0.79°C, i.e., smaller than 1°C (Figure 3), whereas the average MAEs and RMSEs for Tmax and Tmin are >1°C (Tmax: average MAE = 1.20, average RMSE = 1.70; Tmin: average MAE = 1.41, average RMSE = 1.92) (Figures S4 and S5). Therefore, the GPR model requires further improvement regarding interpolation of Tmax and Tmin.

## 6 Data availability

The GPRChinaTemp1km dataset includes monthly maximum air temperature, minimum air temperature, and mean air temperature at 1 km spatial resolution over China from January 1951 to December 2020. The datasets are publicly available in GeoTIFF format on Zenodo at https://doi.org/10.5281/zenodo.5112122 (He et al., 2021a) for monthly maximum air temperature, at https://doi.org/10.5281/zenodo.5111989 (He et al., 2021b) for monthly mean air temperature, and at https://doi.org/10.5281/zenodo.5112232 (He et al., 2021c) for monthly minimum air temperature. The unit of the data is °C.

## 7 Conclusions

A long-term, high-resolution, current, and spatially continuous dataset of air temperature over China is fundamental for understanding climatic dynamics and conducting related scientific research. We used meteorological station data available from January 1951 to December 2020 throughout China as the dependent variable, and longitude, latitude, and elevation were considered as independent variables for interpolation. We used three machine learning models (i.e., RF, SVM, and GPR) to

investigate the potential of machine learning techniques regarding interpolation of air temperature over China. Results showed that GPR performed best, followed by SVM and RF. The machine learning models were also compared with conventional

interpolation methods (i.e., IDW, OK, and ANUSPLIN), and the results showed that GPR was generally superior for interpolating Tmax, Tmin, and Tmean for each month over China. Comparison of the GPR-derived results with existing products (i.e., TerraClimate, FLDAS, and ERA5) revealed that GPR outperformed the three products with regard to Tmax, Tmin, and Tmean. We constructed a new 1 km resolution monthly maximum, minimum, and mean air temperature dataset (named GPRChinaTemp1km) for China from 1951 to 2020 using the advanced GPR machine learning method. Most regions

of China display significant increases for Tmean and Tmin in each month, while the trends of significant increase, non-significant increase, and non-significant decrease are prominent for Tmax. More profound analysis can be conducted based on our temperature datasets, which could help further understanding regarding global warming and climate change.

**Author contributions**

QH and MW designed the research and developed the methodology; QH wrote the manuscript; all other authors reviewed and revised the manuscript.

**Competing interests**

The authors declare that they have no conflict of interest.


**Acknowledgements**

We thank the China Meteorological Data Service Centre for providing the required temperature data of the meteorological stations. We also gratefully acknowledge other institutions and researchers that have contributed to the study. This work was supported by the Research on Key Technologies of Multi-level Accurate Rescue for Major Natural Disasters (grant no.

2017YFC1502902).

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
