# Peer review of "GPRChinaTemp1km: a high-resolution monthly air temperature dataset for China (1951–2020) based on machine learning"

_Earth System Science Data, 2021_

## Referee Comment (RC1)

**Review on 'GPRChinaTemp1km: a high-resolution monthly air temperature dataset for China (1951-2020) based on machine learning' by He et al., 2021.**

The presented study aims to produce a long term, gridded product for monthly air temperature over China at high spatial resolution (1km grid spacing) based on observational data from meteorological stations operated by the China Meteorological Service. This is a challenging task since observations are scattered irregularly over the country with very limited number of station sites in Western China where spatial variability is large due too complex terrain.

Despite the appealing aims of this work which fit well in the scope of the journal, the applied methods show up with major deficiencies. The aspects are listed in the following together with some recommendations for potential improvement. Note that a couple of points have already been mentioned by the referees of the previous submission (see here).

(1) In the introduction, the advantages and disadvantages of different data sources for generating long term, gridded 2m temperature products for China are discussed. However, the inclusion of reanalysis data such as the ERA5 dataset is not considered, even though they provide consistent, dynamical information on the atmospheric state over several decades. Likewise, other relevant studies with similar aims are not mentioned as well, e.g. Peng et al., 2019 [1]

(2) The method for data splitting (random spatial split) is improper since there is strong spatial autocorrelation in the data. Especially, for flat areas with a dense observational network, this leads to an over-simplification for the interpolation task: Even with a naïve mapping to a neighboring station from the training dataset during inference (i.e. corresponding to a nearest neighbor approach), the result would be very good. In other words, the ML-model does not learn from the features and becomes incapable to generalize. Consequently, the results get much poorer, also seen from the much large residuals in regions with complex terrain and a less dense observational network (i.e. in the Himalaya region west of 100°E and south of 40°N). To avoid strong autocorrelation, data must be split along the temporal axis, i.e. by assigning sequential years of data to the test dataset, see, e.g., Section 2.2. in Kleinert et al., 2020 [2] for a more detailed discussion. Note that this aspect is also strongly related to the subsequent major points.

(3) Only static feature variables are used. The only way for the model to get dynamical information is via optimization on the predictand (the 2m observations from the training dataset). This also reasons why a model has to trained for each month of the considered period, resulting in an enormous number of models. Including (coarser) reanalysis data from a numerical atmospheric model to the predictors is therefore strongly recommended, since it allows to circumvent the spatial auto correlation problem (see aspect 2) and to generalize better. Besides, not only elevation is crucial for the near surface temperature, but also strongly the ambient topography (i.e. location in a valley or on a mountain). With the current approach, this information is missing, even though other studies point out its importance (e.g. Sha et al., 2020 [3]).
Furthermore, there are longer periods where some of the predictors are practically uncorrelated with the target quantity. Thus, it's unlikely that these variables contribute to the interpolation model for the respective months (e.g. longitude and elevation for winter months).

(4) Evaluation of the model performance does not serve the aims of the paper, i.e. provide accurate data for regions with sparse observations. Due to the dominance of stations in the flat, densely observed parts of China, the accuracy metrics presented are too optimistic for the regions where the underlying terrain is complex and/or observations are sparse. There are only twenty stations in the Himalaya region (west of 100°E and south of 40°N), thus the results are dominated by the majority of stations located in the flat regions to the east. Alternative data splitting with an adaption of the approach (inclusion of dynamic predictors) would increase the robustness in the evaluation results for the mountainous regions.

(5) There are follow-up deficiencies in the analysis:

(a) The patterns described in Section 4.2. are large-scale patterns that does not relate to the high spatial resolutions of the dataset. The described pattern would be indeed evident in datasets with much coarser resolution, while interesting patterns due to variations in topography/ land use are not investigated.

(b) The trend analysis shows up with patterns in North-Western China for winter months that look like artefacts. The bulls-eye structure as well as the trend gradient are most likely artefacts since observational data are sparse or not existent in this region. At least, references to other work which explicitly deal with trends in the Xinjiang region would be required in addition to a more elaborated and robust method.

(c) ANUSPLIN performs better for July in the 70s, 80s and 90s (see Fig. 8). However, this is not discussed/mentioned subsequently.

(d) Discussion on the limitations does not focus on the real issues with the approach, especially regarding the static predictors.

(6) The comparison with other datasets is misleading. ERA5 and FLADS have a much coarser spatial resolution than the analyzed 1x1 km-dataset (factor of 10 and more). Thus, comparison is not straightforward and at minimum requires a reflection due to mismatches in surface elevation. Rather a comparison to other datasets with matching spatial resolution as presented in [1] would be fair and supportive.

Besides, there are couple of minor comments/issues. The most relevant are listed subsequently:

* Slitting up the data for Tmean, Tmax and Tmin into three datasets is unnecessary. Since the approach is the same, provide in a joint dataset with only one DOI.

* Dynamical and statistical downscaling techniques with reference to reanalysis datasets must be mentioned  in the introduction, e.g. something like COSMO-REA2 (see,e.g., [4]) for other regions on Earth.

* Provide references to the statement on the rather poor quality of remote sensing data (see l.66)

* How is the STRM DEM data remapped onto the 1x1 km-grid? Note: should not be a bilinear, but rather an averaging method.

* Only mentioned used software once. Repeated reference to MATLAB is unnecessary. Rather spent some additional words on the technique itself.

* l.149: Should be 'ensemble machine learning'

* l.219: Unnecessary reference to Equation 4 which directly follows the sentence.

* l.343.f.: Sentence is barely understandable and requires reformulation.

Literature reference:
[1] Peng, Shouzhang, et al. "1 km monthly temperature and precipitation dataset for China from 1901 to 2017." *Earth System Science Data* 11.4 (2019): 1931-1946. DOI.

[2] Kleinert, Felix, Martin G. Schultz, and Lukas H. Leufen. "IntelliO3-ts v1. 0: A neural network approach to predict near-surface ozone concentrations in Germany." *Geoscientific model development discussions* 2020.FZJ-2020-05012 (2020): 1-69. DOI.

[3] Sha, Yingkai, et al. "Deep-learning-based gridded downscaling of surface meteorological variables in complex terrain. Part I: Daily maximum and minimum 2-m temperature." *Journal of Applied [3] Meteorology and Climatology* 59.12 (2020): 2057-2073. DOI.

[4] Wahl, Sabrina, et al. "A novel convective-scale regional reanalysis COSMO-REA2: Improving the representation of precipitation." *Meteorol. Z* 26.4 (2017): 345-361. DOI.

---

## Author Comment (AC2)

Dear Reviewer,

The comments offered have been immensely helpful. We appreciate your insightful comments on our paper. We have responded to every question, indicating exactly how we addressed each concern.

The manuscript aims to produce a long term dataset of monthly 2m temperature over China at high spatial resolution on a 1x1 km grid. While the objective is appealing due to the challenges related to the complex topography and the irregular data availability in the target region, the applied methods show up with significant issues. The major issues are listed subsequently:

Reply:Many thanks for the comments. We apologize for not expressing ourselves clearly. The method we designed generates highly accurate data products. Test results from meteorological observation sites in the field show that our method is robust and repeatable. We have responded to every question to make the expression clearer and more accurate. The point-to-point responses to the comments are listed below.

[1] The introduction discusses advantages and disadvantages of different information sources for the targeted dataset. While strong arguments for point-wise observational data are presented, long term reanalysis data products are not considered despite they provide consistent and spatio-temporally coherent information on the atmospheric state. It is unclear why such data is not considered to provide predictor variables.

Reply: The reanalysis data have some limitations as the predictor variables.

(1) First, the resolution of the reanalysis data is usually low (e.g. the resolution of ERA5 data is 0.25˚). Since the spatial resolution in our study is 1-km, the reanalysis products cannot provide such fine resolution data.
(2) Second, the time span of the reanalysis data can not meet the study period in our study. The period of ERA5 starts from 1979 (Tang et al., 2020) while the dataset we produced starts from 1951.
(3) Third, the reanalysis data are generated using the station observed data, which have uncertainty per se. As shown in the study of Tang (2020), the accuracy of ERA5 data in China is relatively low. The satellite-based and atmospheric reanalysis precipitation estimates are highly constrained by errors (Yin et al., 2021).
(4) The model designed in our study can generate high-resolution datasets without using the reanalysis data.

Considering the above, we did not consider the reanalysis data in our study.

[2] The method of data splitting leads to strong autocorrelation between the training and test dataset. Due to the spatial proximity of stations in both dataset, a fundamental requirement is hurt, that is the independency (or at least a minimization of dependency) between the training and test dataset. This is especially true for the stations located in the flat eastern parts of China with a dense observational network. Thus, the statistical model are prone to learn nearest neighbor-relations

rather than learning real abstractions from the features, see, e.g. Kleinert et al., 2021 for a more detailed discussion on the requirement of splitting the test and training data temporally when stations are located close to each other.

Reply: Many thanks for your constructive comments. In machine learning, there are two strategies for splitting the training set and testing set. The first is a spatial division which split the data into the training set and testing set on the spatial field. The second is temporal split which splits the data into non-overlapping time periods for training and testing, e.g. the study you mentioned (Kleinert et al., 2021). However, there is no standard method for splitting the training and testing dataset. In our study, we used the first strategy for splitting the data, mainly because the following reasons:
(1) The temporal splitting is not appropriate in our study. In the study of Kleinert (Kleinert et al., 2021), they used all the data from 1 January 1997 to 31 December 2007 as the training dataset while it is not feasible in our study to use all the historical data as training sets. Our object is to generate the long time-series data for each month ranging from 1951 to 2020. Thus we need a testing dataset for each month to evaluate the monthly data. In our study, the spatial splitting method can meet the requirements of our study goal.

(2) In the spatial prediction of the environmental variables, numerous study uses the spatial split (Costache et al., 2020; Band et al., 2020; Mohajane et al., 2021; Kutlug Sahin and Colkesen, 2021; Hijmans et al., 2005; Fick and Hijmans, 2017).

(3) The spatial splitting was completed using the "Subset Features" (Geostatistical Analyst) tool in ArcGIS which divides the original dataset into two parts: one part can be used to construct the model; the other part can be used to compare and validate the output. "Subset Features" is the most rigorous way to assess the quality of an output surface. Several studies have used the Subset Features tool of ArcGIS for splitting training and testing datasets in machine learning modelling (Costache et al., 2020; Band et al., 2020; Mohajane et al., 2021; Kutlug Sahin and Colkesen, 2021).

The Subset Features tool divides the data into two subsets. Subset one will have L features, and subset two will have N - L features (with N being the amount of features in the original dataset). The features are divided by generating random values from a uniform [0,1] distribution. If the random value is less than L/N, the feature is assigned to the first subset. If not, the feature is assigned to the second subset. (source: https://desktop.arcgis.com/en/arcmap/latest/extensions/geostatistical-analyst/how-subset-features-works.htm).

(4) We used the 10-fold cross-validation when training the models (Line 193 in the manuscript).

(5) The spatial distribution of the testing data is similar to the spatial distribution of all the data, which is in conformity with the stratified sampling scheme in the machine learning.
We will discuss the splitting strategies in the discussion part of the manuscript.

[3] Only static features are used as predictors which implies that a model must trained for each month (!) of the period under consideration. Thus, dynamic information on the atmospheric state can exclusively deduced from the optimization procedure on the predictand. It is strongly

recommended to introduce dynamical data as a predictor variable instead. Besides, the chosen predictors have periods with neglectable correlation with respect to the target quantity and important features such as the ambient topography (is the meteorological station located in a valley) is absent (see, e.g., Sha et al., 2020).

**Reply:** Many thanks for your comments.

(1) In our study, the model was trained for each month. We described the model construction in Lines 193-194. The longitude, latitude and elevation are indeed static factors, but we construct the model for each month, respectively, which can reflect the changes of temperature from month to month.

(2) The remote sensing data such as NDVI, land use change and surface temperature are usually not available before 2000 since our data is from 1951 to 2020. Furthermore, the MODIS data are not available for each month from January 2000 to December 2020. As shown in Figure 1, the percentage of the available MODIS images are low in northeast China and southern areas. So the remote sensing data are not appropriate for generating long-term temperature data in our study.

(3) Furthermore, there is inherent data inaccuracy in the remote sensing data itself, such as the land use data.

(4) As shown in the study of Sha et al. (2020), the orography, as represented by elevation fields can help characterize the spatial heterogeneity of 2-m temperature. The meteorological processes are locally embedded with small-scale terrain features. The terrain features including plain, slope, peak and valley are recognized as the semantic contents of terrain. They used the elevation data to represent those terrain semantics. In our study, we used the DEM data as the predictor in the machine learning model. As said in the study of Sha et al. (2020), the terrain semantics can be learned from gridded elevation inputs.

(5) We can obtain the high-resolution dataset using the selected predictors in our study. The accuracy evaluation shows the rationality of the predictors. The model is robust in generating the long-term temperature datasets.

[Figure]

Figure 1 Spatial distribution of the percentage of the available MODIS images in each year (2000 - 2020) by excluding clouds.

[4] The evaluation does not serve the objectives of the study. The stations in the test dataset are dominated by stations over flat terrain with a dense observational network. Thus, potential deficiencies in capturing the variations due to underlying complex topography are hidden. Indeed,

Figure 5 indicates that residuals are considerably larger over the mountainous region.

Reply: We agree with the comment. The meteorological stations in mainland China are unevenly distributed with more stations in the flat terrain and less stations in the mountains. This is the inherent data limitation for modelling continuous raster products in China (Guo et al., 2020; Liu et al., 2018). It is true that the stations in the Qinghai-Tibet plateau are sparse (Xu et al., 2018; Zhang et al., 2016). It is also an existing challenge of the spatial interpolation of temperature using the station data. The potential deficiency in capturing the variations in regions with complex terrain is an existing issue in the current studies. We are working to improve the accuracy of models in complex regions. The altitude information is conducive to the estimation of temperature (Berndt and Haberlandt, 2018).

To show the strength of our data in regions with complicated topography. We took the Tibetan plateau region as an example. We compared the accuracy of our data with Peng's data (Peng et al., 2019) in Tibetan plateau. In our study, the accuracy in the Qinghai-Tibet Plateau is relatively good. We used the mean temperature data of Peng et al. (Peng et al., 2019) to make a comparison. The testing stations which were not used in the model training were used to make the comparison. As shown in Figure 2, the RMSE of most months for GPR is lower than Peng and GPR has smaller variation in RMSE. The $R^2$ of GPR shows good accuracy from January to December, with smaller variation in each month, while for Peng's data the variation in summer is quite high.

[Figure]

Figure 2 Comparison between the GPR data in our study and the Peng's data in the Tibetan region

[5] Several issues in the follow-up study are present such as (a) a focus on large-scale temperature patterns instead of fine-scale patterns in Section 4.2. to reason the high spatial resolution of the dataset, (b) the interpretation of patterns in the Xinjiang region which look like artefacts (bulls-eye pattern in winter months) and (c) the missing notification on the better performance of the reference method ANUSPLIN for July-months in the 70s, 80s and 90s.

Reply: The spatial resolution of the temperature dataset in our study is 1 km. Since the territory of China is large, it is challenging to produce fine-scale temperature data. Besides, the scale of 1 km is the resolution of a lot of high-resolution datasets for mainland China, like the 1 km monthly temperature and precipitation dataset (Peng et al., 2019), 1 km daily surface air temperature product over mainland China (Chen et al., 2021), a high-resolution crop phenological dataset for three staple

crops in China (Luo et al., 2020). The 1-km resolution is high enough for mainland China which can satisfy a lot of the requirements in other scientific research or practical applications. The 5-km spatial resolution dataset for Spain which is way smaller than China is also treated as the high-resolution dataset (Serrano-Notivoli et al., 2019). We admit that the finer resolution data may provide more detailed information but the 1-km resolution data is high enough for multiple studies.

(b) The bulls-eye pattern in the Xinjiang region in the winter months is induced by the complex topography, which just shows that the model captured the detailed local differentiation of temperature due to topographic conditions. As shown in Figure 3, the region which has a relatively lower temperature (Section 4.2 Figure 6) is because of the high altitude.

(c) Considering the proven power of ANUSPLIN in predicting meteorological variables, the GPR yields relatively satisfactory results. The accuracy of ANUSPLIN for July-months in the 70s, 80s and 90s is slightly higher than GPR while the accuracy of GPR still performs relatively well. ANUSPLIN uses the thin-plate smoothing spline algorithm which allows the introduction of multivariate linear sub-models with complex model coefficients to be calculated. Using the same computational resources, ANUSPLIN is more time-consuming than GPR. Besides, GPR has higher accuracy in the winter months. We will discuss more in the discussion section.

[Figure]

Figure 3 The elevation of Xinjiang Uygur Autonomous Region

[6] The comparison to the competing datasets ERA5 and FLADS is misleading due to the much coarser spatial resolution of these two datasets. A fair comparison would consult datasets with similar spatial resolution such as the dataset described in Peng et al., 2019.

Reply: Many thanks for your comments. We use three datasets ERA5, FLDAS and TerraClimate. The spatial resolution of the three datasets is 27830 meters, 11132 meters, and 4638.3 meters, respectively. Our datasets are 1-km. We resampled all the data to 27830 meters to keep the resolution consistent and then we comparisons. As shown in Figure 4, GPR still outperformed other products.

[Figure]

Figure 4 Taylor diagrams displaying a statistical comparison with observations between our products generated using the GPR model and the other products under the same spatial resolution.

Besides, we also compared our datasets with Peng's data as you suggested. As shown in Figure 5, our datasets have relatively higher accuracy than Peng's data on the whole, especially in warm months.

[Figure]

Figure 5 Accuracy comparison between the GPR data and the Peng's data for mean temperature

Further minor issues are:

* Splitting into three distinct datasets is unnecessary. Rather merge it to one dataset with one DOI.
Reply: The Zenodo database has limitations for the data size (max 50 GB per dataset). The zip format file of each dataset is about 30 GB, so we uploaded mean, maximum and minimum temperature data, respectively.

* Refer to statistical and dynamical downscaling techniques in the introduction.
Reply: Thanks for your suggestion. The downscaling technique uses the existing coarse product to produce the high-resolution dataset. The interpolation uses the meteorological station data to generate the spatial continuous grid dataset. The two strategies use different source data, but they have the same objectives. In fact, there are multiple low spatial resolution datasets, such as the Climatic Research Unit (CRU) (Harris et al., 2014), the Global Precipitation Climatology Centre (GPCC) (Schneider et al., 2014; Becker et al., 2013), and Willmott & Matsuura (W&M) (Matsuura and Willmott, 2012) which are generated using the data from the observational stations. It is a reliable way to produce continuous datasets using the observed station data (Peng et al.,

2019).

* Provide references to the problems related to remote sensing data (see l.66).
Reply: Thanks for your reminder. The references are listed below:
(Dong and Xiao, 2016)
(Xiao et al., 2018, p.2013–2016)
(Mao et al., 2019)

* Describe the remapping of the STRM DEM data onto the 1x1 km grid (should be an averaging method).
Reply: We used GEE to export the STRM DEM data as 1*1 km grid. The "Scale" parameter was used to specify the output resolution to 1 km. The concept of "Scale" is illustrated in Figure 6 (https://developers.google.com/earth-engine/guides/scale). The default method of resampling is the nearest neighbour (https://developers.google.com/earth-engine/guides/scale#image-pyramids).

[Figure]

Figure 6 A graphic representation of an image dataset in Earth Engine. Dashed lines represent the pyramiding policy for aggregating 2x2 blocks of 4 pixels. Earth Engine uses the scale specified by the output to determine the appropriate level of the image pyramid to use as input.

* The used software tool MATLAB should be only mentioned once rather than being repeated three times. More details on the respective ML-technique would be appreciated.
Reply: Thanks a lot for your advice. We should mention the MATLAB once. In the light of the limitation of the words in the manuscript, we did not provide so many details for all the machine learning methods but we provided the references or related links which have detailed descriptions of the machine learning methods. In the next revision, we will add more key information about the machine learning methods in the manuscript.

* l.149: Should be 'ensemble machine learning'
Reply: You are right. Thanks for pointing this out and sorry for the wrong spelling.

* l.219: "Unnecessary reference to Equation 4 which directly follows the sentence.
Reply: Thanks for your comment. The references should be removed.

* l.343f. This is sentence is barely comprehensible.
Reply: What we want to express is that the accuracy measures fluctuate with the seasons.

**References**

Subset Features (Geostatistical Analyst)—ArcGIS Pro | Documentation: https://pro.arcgis.com/en/pro-app/latest/tool-reference/geostatistical-analyst/subset-features.htm, last access: 28 January 2022.

Band, S. S., Janizadeh, S., Chandra Pal, S., Saha, A., Chakrabortty, R., Melesse, A. M., and Mosavi, A.: Flash Flood Susceptibility Modeling Using New Approaches of Hybrid and Ensemble Tree-Based Machine Learning Algorithms, 12, 3568, https://doi.org/10.3390/rs12213568, 2020.

Becker, A., Finger, P., and Meyer-Christo, A.: A description of the global land-surface precipitation data products of the Global Precipitation Climatology Centre with sample applications including centennial (trend) analysis from 1901–present, 29, https://doi.org/10.5194/essd-5-71-2013, 2013.

Berndt, C. and Haberlandt, U.: Spatial interpolation of climate variables in Northern Germany—Influence of temporal resolution and network density, Journal of Hydrology: Regional Studies, 15, 184–202, https://doi.org/10.1016/j.ejrh.2018.02.002, 2018.

Chen, Y., Liang, S., Ma, H., Li, B., He, T., and Wang, Q.: An all-sky 1 km daily surface air temperature product over mainlandChina for 2003–2019 from MODIS and ancillary data, Data, Algorithms, and Models, https://doi.org/10.5194/essd-2021-31, 2021.

Costache, R., Pham, Q. B., Sharifi, E., Linh, N. T. T., Abba, S. I., Vojtek, M., Vojteková, J., Nhi, P. T. T., and Khoi, D. N.: Flash-Flood Susceptibility Assessment Using Multi-Criteria Decision Making and Machine Learning Supported by Remote Sensing and GIS Techniques, 12, 106, https://doi.org/10.3390/rs12010106, 2020.

Dong, J. and Xiao, X.: Evolution of regional to global paddy rice mapping methods: A review, ISPRS Journal of Photogrammetry and Remote Sensing, 119, 214–227, https://doi.org/10.1016/j.isprsjprs.2016.05.010, 2016.

Fick, S. E. and Hijmans, R. J.: WorldClim 2: new 1-km spatial resolution climate surfaces for global land areas, 37, 4302–4315, https://doi.org/10.1002/joc.5086, 2017.

Guo, B., Zhang, J., Meng, X., Xu, T., and Song, Y.: Long-term spatio-temporal precipitation variations in China with precipitation surface interpolated by ANUSPLIN, Sci Rep, 10, 81, https://doi.org/10.1038/s41598-019-57078-3, 2020.

Harris, I., Jones, P. d., Osborn, T. j., and Lister, D. h.: Updated high-resolution grids of monthly climatic observations – the CRU TS3.10 Dataset, 34, 623–642, https://doi.org/10.1002/joc.3711, 2014.

Hijmans, R. J., Cameron, S. E., Parra, J. L., Jones, P. G., and Jarvis, A.: Very high resolution interpolated climate surfaces for global land areas, Int. J. Climatol., 25, 1965–1978, https://doi.org/10.1002/joc.1276, 2005.

Kleinert, F., Leufen, L. H., and Schultz, M. G.: IntelliO3-ts v1.0: a neural network approach to predict near-surface ozone concentrations in Germany, Geosci. Model Dev., 14, 1–25, https://doi.org/10.5194/gmd-14-1-2021, 2021.

Kutlug Sahin, E. and Colkesen, I.: Performance analysis of advanced decision tree-based ensemble learning algorithms for landslide susceptibility mapping, 36, 1253–1275, https://doi.org/10.1080/10106049.2019.1641560, 2021.

Liu, Z., Liu, Y., Wang, S., Yang, X., Wang, L., Baig, M. H. A., Chi, W., and Wang, Z.: Evaluation of

Spatial and Temporal Performances of ERA-Interim Precipitation and Temperature in Mainland China, 31, 4347–4365, https://doi.org/10.1175/JCLI-D-17-0212.1, 2018.

Luo, Y., Zhang, Z., Chen, Y., Li, Z., and Tao, F.: ChinaCropPhen1km: a high-resolution crop phenological dataset for three staple crops in China during 2000–2015 based on leaf area index (LAI) products, 12, 197–214, https://doi.org/10.5194/essd-12-197-2020, 2020.

Mao, K., Yuan, Z., Zuo, Z., Xu, T., Shen, X., and Gao, C.: Changes in Global Cloud Cover Based on Remote Sensing Data from 2003 to 2012, Chin. Geogr. Sci., 29, 306–315, https://doi.org/10.1007/s11769-019-1030-6, 2019.

Matsuura, K. and Willmott, C. J.: Terrestrial precipitation: 1900-2010 gridded monthly time series, University of Delaware, 2012.

Mohajane, M., Costache, R., Karimi, F., Bao Pham, Q., Essahlaoui, A., Nguyen, H., Laneve, G., and Oudija, F.: Application of remote sensing and machine learning algorithms for forest fire mapping in a Mediterranean area, Ecological Indicators, 129, 107869, https://doi.org/10.1016/j.ecolind.2021.107869, 2021.

Peng, S., Ding, Y., Liu, W., and Li, Z.: 1 km monthly temperature and precipitation dataset for China from 1901 to 2017, Earth Syst. Sci. Data, 11, 1931–1946, https://doi.org/10.5194/essd-11-1931-2019, 2019.

Schneider, U., Becker, A., Finger, P., Meyer-Christoffer, A., Ziese, M., and Rudolf, B.: GPCC's new land surface precipitation climatology based on quality-controlled in situ data and its role in quantifying the global water cycle, Theor Appl Climatol, 115, 15–40, https://doi.org/10.1007/s00704-013-0860-x, 2014.

Serrano-Notivoli, R., Beguería, S., and de Luis, M.: STEAD: a high-resolution daily gridded temperature dataset for Spain, 18, 2019.

Tang, G., Clark, M. P., Papalexiou, S. M., Ma, Z., and Hong, Y.: Have satellite precipitation products improved over last two decades? A comprehensive comparison of GPM IMERG with nine satellite and reanalysis datasets, Remote Sensing of Environment, 240, 111697, https://doi.org/10.1016/j.rse.2020.111697, 2020.

Xiao, C., Li, P., Feng, Z., and Wu, X.: Spatio-temporal differences in cloud cover of Landsat-8 OLI observations across China during 2013–2016, Journal of Geographical Sciences, 28, 429–444, https://doi.org/10.1007/s11442-018-1482-0, 2018.

Xu, Y., Knudby, A., Shen, Y., and Liu, Y.: Mapping Monthly Air Temperature in the Tibetan Plateau From MODIS Data Based on Machine Learning Methods, 11, 345–354, https://doi.org/10.1109/JSTARS.2017.2787191, 2018.

Yin, J., Guo, S., Gu, L., Zeng, Z., Liu, D., Chen, J., Shen, Y., and Xu, C.-Y.: Blending multi-satellite, atmospheric reanalysis and gauge precipitation products to facilitate hydrological modelling, Journal of Hydrology, 593, 125878, https://doi.org/10.1016/j.jhydrol.2020.125878, 2021.

Zhang, H., Zhang, F., Ye, M., Che, T., and Zhang, G.: Estimating daily air temperatures over the Tibetan Plateau by dynamically integrating MODIS LST data, 121, 11,425-11,441, https://doi.org/10.1002/2016JD025154, 2016.

---

## Author Comment (AC3)

Dear Reviewer,

The comments offered have been immensely helpful. We appreciate your insightful comments on our paper. We have responded to every question, indicating exactly how we addressed each concern.

**General Comments:**

The paper is presenting a method for spatial interpolation of air temperature data for China from meteorological stations based on machine learning tools. The authors analyze the technique used and present the limitations of the experiment. Three ML models were tested and three interpolation method and the Gaussian Process Regression was chosen based on its better performance. The results compared with existing published datasets. A detailed trend analysis of the predicted dataset is also presented. ML techniques are very promising as they are addressing current challenges in computational research.

Response: Many thanks for the constructive comments on our study.

**Specific comments:**

Q1: It is not clear to me if all stations contribute equally to the analysis, for example red and blue stations (Figure S2). Is there any weighting technique applied to the training model(s)? If yes, I think it could be mentioned.

Response: In our study, we used the "subset features" option of the Geostatistical Analyst Tools in ArcGIS10.8 to divide the original dataset into 70% training dataset and 30% testing dataset. This tool considers the randomness both in the data and the spatial distribution of the data. There is no weighting technique applied in the training models.

Q3: Besides the trends statistical analysis, are there available error spatial distributions of the predicted temperatures so as to illustrate the confidence level of the analysis results especially in station empty areas?

Response: GPR is a full Bayesian learning algorithm. A process is referred to as a Gaussian process if it is assumed that the joint probability distribution of model outputs is Gaussian (Zhu et al., 2018). Because a GPR model is probabilistic, it is possible to compute the prediction intervals using the trained model. As you suggested, we will provide the spatial confidence level graph for each month (with a significance level of 95%). Here, we displayed the spatial distribution of the width of the predicted intervals with a significance level of 95% ( using the upper limit minus the lower limit) for the 12 months in 2010 using the trained models (Figure 1). Figure 2 provides the histogram of the width of the confidence intervals. The calculation was conducted in Matlab 2021b. For more detailed information about the prediction intervals of GPR models, please see https://www.mathworks.com/help/stats/gaussian-process-regression-models.html.

[Figure]

Figure 1 The spatial distribution of the width of the 95% prediction intervals (the upper limit minus the lower limit) for 12 months in 2010

[Figure]

Figure 2 The histogram of the width of the 95% prediction intervals (the upper limit minus the lower limit) for 12 months in 2010

Q4: In addition, as the study of the climatic dynamics is in the epicenter of this work, the use of remote sensing data jointly with the land meteo stations could overcome the data scarcity, improve the results and reveal trends with more accuracy after 2000.

Response: The goal of our study is to generate the long-term time series of high-resolution temperature data. Due to the limitation of the remote sensing data, we did not consider remote sensing data in our study. We discussed the topic in the discussion session.

The longitude, latitude and elevation are static factors, but we construct the model for each month, respectively, which can reflect the changes of temperature from month to month. The remote sensing data such as NDVI, land-use change and surface temperature are usually not available before 2000 since our data is from 1951 to 2020. Furthermore, the MODIS data are not available for each month from January 2000 to December 2020. As shown in Figure 3, the percentage of the available MODIS images are low in northeast China and southern areas. So we did not use the remote sensing data for generating long-term temperature data in our study.

We will consider using the remote sensing data in future studies to further increase the accuracy.

[Figure]

Figure 3 Spatial distribution of the percentage of the available MODIS images in each year (2000 - 2020) by excluding clouds.

Q5: While the height of the air Temperature is mentioned for the ERA5 dataset (2m), this is not the case for the GPRChinaTemp1km product or the other datasets mentioned in the analysis.

Response: We will include more detailed descriptions of the data we used. In our study, we use the weather station data to interpolate the gridded temperature datasets. The station data used in the study records the temperature at 2 m height above ground (Liu et al., 2011; Zhang et al., 2010). Thus, the generated GPRChinaTemp1km product also represents the temperature data at 2 m height.

For TerraClimate data, it is produced based on other datasets including WorldClim, CRUTs4.0 and JRA-55 (Abatzoglou et al., 2018, p.1958–2015). The temperatures in WorldClim are at 2 m height (Fick and Hijmans, 2017; Chou et al., 2020). The temperature from CRU Ts and JRA-55 are also at 2 m height (Harris et al., 2020; https://jra.kishou.go.jp/JRA-55/document/JRA-55_handbook_LL125_en.pdf). Therefore, the TerraClimate dataset also represents the 2m temperature.

The height of the temperature data from FLDAS is also 2 m (McNally et al., 2017; https://ldas.gsfc.nasa.gov/fldas/specifications).

We will include more detailed information on the height of the data we used in the revised manuscript.

Q6: Could the experiment be tested in other atmospheric parameters? If so, I think that a few sentences on the perspectives of the specific approach would be beneficial.

Response: Thanks a lot for your suggestion. Our current study is only aiming for temperature. We have not done some experiments on other meteorological variables. This actually is our next work. We are trying to apply the GPR model to other meteorological variables. We will add a few sentences to describe that in the Discussion session.

**References:**

Abatzoglou, J. T., Dobrowski, S. Z., Parks, S. A., and Hegewisch, K. C.: TerraClimate, a high-resolution global dataset of monthly climate and climatic water balance from 1958–2015, 5, 170191, https://doi.org/10.1038/sdata.2017.191, 2018.

Chou, S. C., de Arruda Lyra, A., Gomes, J. L., Rodriguez, D. A., Alves Martins, M., Costa Resende, N., da Silva Tavares, P., Pereira Dereczynski, C., Lopes Pilotto, I., Martins, A. M., Alves de Carvalho, L. F., Lima Onofre, J. L., Major, I., Penhor, M., and Santana, A.: Downscaling projections of climate change in Sao Tome and Principe Islands, Africa, Clim Dyn, 54, 4021–4042, https://doi.org/10.1007/s00382-020-05212-7, 2020.

Fick, S. E. and Hijmans, R. J.: WorldClim 2: new 1-km spatial resolution climate surfaces for global land areas, 37, 4302–4315, https://doi.org/10.1002/joc.5086, 2017.

Harris, I., Osborn, T. J., Jones, P., and Lister, D.: Version 4 of the CRU TS monthly high-resolution gridded multivariate climate dataset, Sci Data, 7, 109, https://doi.org/10.1038/s41597-020-0453-3, 2020.

Liu, X., Luo, Y., Zhang, D., Zhang, M., and Liu, C.: Recent changes in pan-evaporation dynamics in China, 38, https://doi.org/10.1029/2011GL047929, 2011.

McNally, A., Arsenault, K., Kumar, S., Shukla, S., Peterson, P., Wang, S., Funk, C., Peters-Lidard, C. D., and Verdin, J. P.: A land data assimilation system for sub-Saharan Africa food and water security applications, Sci Data, 4, 170012, https://doi.org/10.1038/sdata.2017.12, 2017.

Zhang, X., Kang, S., Zhang, L., and Liu, J.: Spatial variation of climatology monthly crop reference evapotranspiration and sensitivity coefficients in Shiyang river basin of northwest China, Agricultural Water Management, 97, 1506–1516, https://doi.org/10.1016/j.agwat.2010.05.004, 2010.

Zhu, S., Nyarko, E. K., and Hadzima-Nyarko, M.: Modelling daily water temperature from air temperature for the Missouri River, PeerJ, 6, e4894, https://doi.org/10.7717/peerj.4894, 2018.

---

## Author Comment (AC4)

Dear Reviewer,

The comments offered have been immensely helpful. We appreciate your insightful comments on our paper. We have responded to every question, indicating exactly how we addressed each concern.

This study describes a new 1km dataset of monthly-mean, monthly-maximum and monthly-minimum surface temperature's over China, developed using machine learning methods. The method used for the final data set was chosen as the best performing method, after a comparison of three modern techniques. A dataset of 613 weather stations over China was used to train and test the machine learning methods. This study is very clearly written and the Figures are of high quality. I agree with all of reviewer 1's comments, so will not repeat these points and assume they have been addressed within the manuscript, but I will add a few further comments below.

Response: Many thanks for the constructive comments.

**General comments:**

Q1: There is always a tradeoff between spatial and temporal resolution when designing new data products. Can you explain in a few sentences in the manuscript why you chose to create a product with such high spatial resolution but such low temporal resolution? You make comparisons at the end to the ERA5 dataset which does have much lower spatial resolution (~30 x less) but it has hourly temporal resolution (720 x more) which is very useful for a number of applications. Comments suggesting the applications where you think this dataset may be preferable to the others mentioned would also be useful.

Response: Thanks a lot for your suggestion. Here are the reasons we produce the monthly product with high spatial resolution. First, the monthly temperature data is crucial for multiple studies and applications such as agriculture (Meshram et al., 2020), meteorological disasters (Tigkas et al., 2019) and ecology (Leihy et al., 2018). Second, the station data we obtained are from the China Meteorological Data Service Centre where the daily temperature data are not available. Thirdly, the daily temperature data with a high spatial resolution for a long period is enormously huge. Creating the data and storing the data for us is still quite challenging. ERA5 has high temporal resolution while the spatial resolution is low. We will mention this in the discussion session in the later revision of our paper.

Q2: You mention ERA5 is only available from 1979, but it is now available back to 1950, so could be used to incorporate dynamical variables (as suggested by reviewer 1). I'm not suggesting you do this, but in the limitations this could be a point for future development. And the text should be updated to reflect the availability of ERA5.

Response: The ERA5 is collected and processed on the Google Earth Engine platform, where the data is only available from 1979. We will update the year in the manuscript. Incorporating the ERA5 can be a good point for our future study. Thanks a lot for your advice. We will add some text to discuss ERA5 in the Discussion section.

Q3: Is any quality control performed on the meteorological station data you use as inputs? A few stations with low quality data could skew the results in data sparse regions.

Response: Yes. We train the model for each month. In each month, we deleted the stations with 999999 or 999998 values which mean the station in that month has no data. We also checked the value range of the air temperature in different months and all the selected stations are with reasonable values.

Q4: Do you know if the final model output is sensitive to the choice of stations used in the test/training dataset? I imagine that this could heavily influence the results in the data sparse regions.

Response: In order to find out if the model result is sensitive to the selection of weather stations used in the training and testing dataset, we conducted some experiments by randomly splitting the data into training and testing sets 50 times. We used the data from 1990, 2000, 2010 to do the case study. As shown in Figure 1, the RMSE varies slightly from different scenarios of the test/training dataset, while there is no obvious variation in $R^2$ (Figure 2). In our study, we split the stations into testing and training stations in ArcGIS, which has considered the spatial distribution of the weather stations. We will put the relationship between the choice of stations with the model output in the Discussion session.

[Figure]

Figure 1 The RMSE using different testing and training datasets

[Figure]

Figure 2 The $R^2$ using different testing and training datasets

Q5: Although you've included elevation, latitude and longitude there are multiple climatic regions in China, and a large amount of external drivers to variations in temperatures. The strength of these may modulate surface temperature behavior (e.g. the strength/location of the monsoon criculation, El Nino southern Oscillation, and other global teleconnections). Distance from the ocean could also play a role. Have you considered these in your explanations for months/stations with particuarly large residuals, or stations with strange behaviors? It could be that if a month had anomalous large scale weather conditions, which your machine learning methods are not trained to capture there are large residuals? These could make interesting case studies and could motivate future work incorporating some dynamical predictors.

Response: Many thanks for your comments. The global teleconnections can influence the surface temperature. This is a good topic for further study. We may use the data we generated combined with global teleconnections to do some research. In order to find out if the generated data in our study can capture the anomalous event, we did a case study in the Sichuan province. In 2006, there is an extremely severe drought event with an extremely high temperature in Sichuan (Li et al., 2011c). We extracted our maximum temperature data to the stations which were not used in the model training. We used nine stations in Sichuan and compared the mean temperature in July from 2003 to 2009. As shown in Figure 3, the temperature in 2006 is markedly higher than the neighbouring years, which means that our data can capture the anomalous condition.

[Figure]

Figure 3 Maximum temperature for 9 different testing stations in Sichuan province

Q6: Figure 2: This is a nice depiction of the relationships. If you could briefly unpack the meteorological understanding behind this in the text it would be beneficial to readers. Have you checked that the relationships hold if different climatic regions of China are subset out?

Response: We used the subregions provided by Zhang et al. (Zhang et al., 2021). The subregions are divided according to the gradients of elevations and the precipitation patterns (Chen and Li, 2016; Zhang et al., 2021). The subregions are shown in Figure 4. We recalculated the correlation coefficients in each sub-region (Figure 5). The results show that Region 4, Region 5, Region 7 and Region 8 have clearly similar relationships as shown in Figure 2 in the manuscript. Since we train the model for the whole area of mainland China, the correlation in different sub-regions would not influence much about the whole region. It is an interesting topic to discuss different sub-regions, which can be helpful for regional studies. We would like to talk about this in the discussion session in our revision.

[Figure]

Figure 4 The division of the subregions, and the spatial distribution of the weather stations over the Chinese mainland.

[Figure]

Figure 5 Correlation coefficients in different subregions

Q7: Line 190-195. So you have 840 different models. Can you comment on how different are all the 70 models for each month? (e.g. do all the January models look very similar?) This could be useful to understand if there are dynamical meteorological explanations for any outliers.

Response: We use the GPR model for all the 840 models for each month. The explicit basis in the GPR model is "constant" and the kernel function of the GPR algorithm is the exponential kernel. The predictor variables were standardized in the GPR. For each month, we use the temperature data of this month to train the model and then use this model to generate the grid data.

As the answer to Question 5: In order to find out if the generated data in our study can capture the anomalous event, we did a case study in the Sichuan province. In 2006, there is an extremely severe drought event with extreme high temperature in Sichuan (Li et al., 2011c). We extracted our maximum temperature data to the stations which were not used in the model training. We used nine stations in Sichuan and compared the mean temperature in July from 2003 to 2009. the temperature in 2006 is markedly higher than the neighboring years (Figure 6), which means that our data can capture the anomalous condition and it can be useful to understand the dynamical meteorological explanations for outliers.

[Figure]

Figure 6 Maximum temperature for 9 different testing stations in Sichuan province

Q8: Line 243: Do you have a sense of why the errors are larger in the colder months? Are the impacts of local meteorological conditions larger in the cold season, which would make it more difficult for the methods to work? There may be meteorological literature on this.

Response: Thanks for your comments. As shown in Figure 2 in the manuscript, there are two variables (i.e., longitude and elevation) that have higher correlation with temperature, while in cold months there is only one variable with relatively higher correlation (i.e., latitude). We did some literature review. The large-scale mountain area is an important source of uncertainty in the temperature mapping, which can influence the spatial distribution of the surface air temperature such as in the area of the Qinghai Tibet Plateau and northwest China (Xu et al., 2018). The study of Stahl et al (Stahl et al., 2006) shows that the standard deviation for daily air temperature in winter is larger than that in summer. The study of Brunetti shows that the interpolation of high-resolution temperature for Italy has the lowest errors in spring and autumn and the highest errors in winter and explained that the elevation coefficients (lapse rates) are markedly different during winter (Brunetti et al., 2014). The air temperature in winter changes rapidly which may be a reason for the high estimation errors in winter (Amini et al., 2019). Rolland (Rolland, 2003) also found higher interpolation reliability for maximum and minimum temperature in winter than in summer. We will add the literature review in the Discussion session.

Q9: Figure 6: Can you comment on any features you're resolving here that are not seen in the lower resolution gridded products you will compare to? There are some very high resolution features on the map, but clarification that they are physical would be useful.

Response: The low spatial resolution can limit the ability to reflect the effects of complex topographies, land surface characteristics, and other processes on climate systems (Peng et al., 2019; Xu et al., 2017). The fine-scale data can provide realistic and reliable climate change information. We displayed the mean temperature of July, 2010 in the same region using ERA, FLADS datasets and the GPR dataset generated in our study (Figure 7). The GPR data can provide more spatial details than ERA and FLDAS (Figure 7). We will add more clarification in our manuscript.

[Figure]

Figure 7 Comparison between the ERA, FLDAS and GPR datasets using the mean air temperature in July 2010

Q10: Does your trend analysis agree with the existing literature on global warming over China? If so include references to this.

Response: Thanks for your suggestion. We did a literature review and found some references which agree with the trend analysis in our study. The distribution of the temperature trend in China in our study agrees with the existing literature (Dong et al., 2015, p.1963–2012; Sun et al., 2018; You et al., 2021; Cui et al., 2017, p.1960–2015)

Q11: Figure 11: The Taylor diagrams show clear improvement from your new dataset. Also including some timeseries from locations not sampled from the observation network compared between the three datasets would be useful to understand how the four products sample the seasonal cycles of the variables.

Response: Many thanks for your advice. We randomly selected three locations not sampled from the observation network to make a comparison. The location of the points is shown in Figure 8. We extract the ERA, FLDAS and the GPR mean air temperature data to the three new points to make a comparison. As shown in Figure 9, the mean air temperature data of the three datasets have the similar pattern from January 1982 to December 2019. We do not have actual temperature data from the three randomly selected points, so it is not possible to make an accuracy comparison. However, the ERA and FLADS datasets are already used in a lot of studies (McNally et al., 2017; Hersbach et al., 2020), which means they are reliable to some extent. Thus the similar pattern between the GPR data and the ERA/FLDAS data can show the reliability of the GPR dataset we generated to a certain extent.

[Figure]

Figure 8 The location of the three random points

[Figure]

Figure 9 Comparisons of the data time series for mean air temperature in three random points

from January 1982 to December 2019

**Small corrections:**

Q12: The height of the air temperatures (surface, 1.5m, 2m) should be added to the manuscript when this is mentioned.

Response: The height of the air temperatures from the weather stations is 2 m. We will add this to section 2.1 Meteorological station data.

Q13: The acronyms for datasets/methods should be defined in the abstract to make it easier to read.

Response: ANUSPLIN: (short for Australian National University Spline); TerraClimate: Monthly Climate and Climatic Water Balance for Global Terrestrial Surfaces; FLDAS: Famine Early Warning Systems Network (FEWS NET) Land Data Assimilation System, and ERA5: ECMWF Climate Reanalysis. We will add the full definition in the Abstract as you suggested.

Q14: Line 38: after commenting on the limitations of the observing stations you could comment here on the limitations of reanalysis based products.

Response: Thanks for the suggestion. The reanalysis based data usually have low spatial resolution, which limits their ability to reflect the effects of complex topographies, land surface characteristics, and other processes on climate systems (Peng et al., 2019; Xu et al., 2017). We will add this comment to our manuscript.

Q15: Throughout the text when you say 'high resolution' this should be changed to 'high spatial resolution' e.g. line 55.

Response: Yes. It should be changed to "high spatial resolution" to be clearer.

Q16: Line 56: 'traditional interpolation techniques' might be clearer?

Response: We agree with this. We will revise it in the manuscript.

Q17: Line 58-60: You comment on a few studies which talk about the superior performance of machine learning techniques but you do not say what the benchmark is that they've succeeded against. This should be included.

Response: The machine learning methods were selected mainly according to the applications of machine learning methods in previous studies. There is potential in applying machine learning methods to predict spatially continuous variables. The combination of machine learning and the traditional model can usually have better performance (Appelhans et al., 2015; Li et al., 2011b). Secondary information considered such as slope, latitude and longitude can improve the performance of machine learning as they provide essential information for machine learning methods. (Li et al., 2011b; Alizamir et al., 2020; Appelhans et al., 2015; Kisi et al., 2017; Zhu et al., 2018). We will include more descriptions in the revised manuscript.

Q18: Line 61: 'estimation of short-term air temperature ' – I'm not sure what you mean by this?

Response: We are sorry for making you confused. Here we should delete "short-term" to make it clearer.

Q19: Line 86: The link here gives me an Error 404.

Response: The link is not available recently, because the data were not available online. We will change our link to the homepage (https://data.cma.cn/data/).

Q20: Around the discussion for Figure 1 it would be interesting to know the spatial distance between observation sites. This might be a small indication of confidence in the final machine learning model output.

Response: Thanks for your advice. We generated the Euclidean distance for the observation stations. Figure 10 shows that the Euclidean distance is quite small in most of the region. The Euclidean distance is relatively large in the west of the Tibetan Plateau and the small region in Inner Mongolia. The larger Euclidean distance means the stations in that region are sparse, which can have an impact on the interpolation accuracy (Hijmans et al., 2005; Li et al., 2011a). We will talk about the model output and the spatial distance in the revised manuscript.

[Figure]

Figure 10 Euclidean distance of the weather stations

Q21: Section 2.3: When commenting on the spatial resolution of the gridded products used for comparison it would be useful to also have this in km.

Response: Thanks for your suggestion. The spatial resolution of TerraClimate, FLDAS, and ERA5 are about 4.6 km, 11 km and 28 km, respectively. We will update the text in the manuscript according to your suggestion.

Q22: Line 149: 'machining learning' should be 'machine learning'

Response: We will change the wrong spelling.

Q23: Section 3.2.2 Are the choices of parameters for the SVM method standard in the literature? Can you please comment on your choices?

Response: Thanks for your suggestion. For the SVM models, we used the Gaussian kernel. The kernel scale is set to 1.7 for all SVM models. Hyperparameters are important for the performance of the model. We optimized the kernel scale of SVM and made a comparison. The box constraint and the epsilon hyperparameters are varying from month to month according to the training data of each month are different. The mean temperature data from January to December from 1951 to 2020 was used for comparing the new model with varying kernel scales and the used model in our study. The accuracy of each month between the optimized model and the used model in the paper is similar (Figure 11), the optimized models do not improve significantly. As we can see that the adjustment of the hyperparameters has little impact on the model accuracy in our study. Besides, the models used in our study are more time-saving and efficient.

[Figure]

Figure 11 Comparison between optimized SVM model (new) and the used model in the paper (old).

Q24: Line 342: ' shows a cyclic pattern' might be clearer.

Response: We will change the text as suggested.

---

## Author Response (AR1)

Dear Editor,

Thank you for your letter and the chance of revising our paper on "GPRChinaTemp1km: a high-resolution monthly air temperature dataset for China (1951–2020) based on machine learning" (Manuscript ID: essd-2021-442). We thank you and the reviewers for giving our manuscript insightful comments to further improve our manuscript.

We have revised our manuscript following your advice. We have included the comments in this letter and responded to them individually. The revisions have been approved by all three authors. The responses to the comments are listed below in blue.

**Reviewer 1**

The comments offered have been immensely helpful. We appreciate your insightful comments on our paper. We have responded to every question, indicating exactly how we addressed each concern.

The manuscript aims to produce a long term dataset of monthly 2m temperature over China at high spatial resolution on a 1x1 km grid. While the objective is appealing due to the challenges related to the complex topography and the irregular data availability in the target region, the applied methods show up with significant issues. The major issues are listed subsequently:

Response: Many thanks for the comments. We apologize for not expressing ourselves clearly. The method we designed generates highly accurate data products. Test results from meteorological observation sites in the field show that our method is robust and repeatable. We have responded to every question to make the expression clearer and more accurate. The point-to-point responses to the comments are listed below.

Q1: The introduction discusses advantages and disadvantages of different information sources for the targeted dataset. While strong arguments for point-wise observational data are presented, long term reanalysis data products are not considered despite they provide consistent and spatio-temporally coherent information on the atmospheric state. It is unclear why such data is not considered to provide predictor variables.

Response: The reanalysis data have some limitations as the predictor variables.

(1) First, the resolution of the reanalysis data is usually low (e.g. the resolution of ERA5 data is 0.25°). Since the spatial resolution in our study is 1-km, the reanalysis products cannot provide such fine resolution data.

(2) Second, the time span of the reanalysis data can not meet the study period in our study. The period of ERA5 starts from 1979 (Tang et al., 2020) on the GEE platform while the dataset we produced starts from 1951.

(3) Third, the reanalysis data are generated using the station observed data, which have uncertainty per se. As shown in the study by Tang (2020), the accuracy of ERA5 data in China is relatively low. The satellite-based and atmospheric reanalysis precipitation estimates are highly constrained by errors (Yin et al., 2021).

(4) The model designed in our study can generate high-resolution datasets without using the reanalysis data.

Considering the above, we did not consider the reanalysis data in our study. We mentioned this in the revised manuscript (**Lines 49-51**).

**Q2: The method of data splitting leads to strong autocorrelation between the training and test dataset. Due to the spatial proximity of stations in both dataset, a fundamental requirement is hurt, that is the independency (or at least a minimization of dependency) between the training and test dataset. This is especially true for the stations located in the flat eastern parts of China with a dense observational network. Thus, the statistical model are prone to learn nearest neighbor-relations rather than learning real abstractions from the features, see, e.g. Kleinert et al., 2021 for a more detailed discussion on the requirement of splitting the test and training data temporally when stations are located close to each other.**

Response: Many thanks for your constructive comments. In machine learning, there are two strategies for splitting the training set and testing set. The first is a spatial division which split the data into the training set and testing set on the spatial field. The second is temporal split which splits the data into non-overlapping time periods for training and testing, e.g. the study you mentioned (Kleinert et al., 2021). However, there is no standard method for splitting the training and testing dataset. In our study, we used the first strategy for splitting the data, mainly because the following reasons:

(1) The temporal splitting is not appropriate in our study. In the study of Kleinert (Kleinert et al., 2021), they used all the data from 1 January 1997 to 31 December 2007 as the training dataset while it is not feasible in our study to use all the historical data as training sets. Our object is to generate the long time-series data for each month ranging from 1951

to 2020. Thus we need a testing dataset for each month to evaluate the monthly data. In our study, the spatial splitting method can meet the requirements of our study goal.

(2) In the spatial prediction of the environmental variables, numerous study uses the spatial split (Costache et al., 2020; Band et al., 2020; Mohajane et al., 2021; Kutlug Sahin and Colkesen, 2021; Hijmans et al., 2005; Fick and Hijmans, 2017).

(3) The spatial splitting was completed using the "Subset Features" (Geostatistical Analyst) tool in ArcGIS which divides the original dataset into two parts: one part can be used to construct the model; the other part can be used to compare and validate the output. "Subset Features" is the most rigorous way to assess the quality of an output surface. Several studies have used the Subset Features tool of ArcGIS for splitting training and testing datasets in machine learning modelling (Costache et al., 2020; Band et al., 2020; Mohajane et al., 2021; Kutlug Sahin and Colkesen, 2021).

The Subset Features tool divides the data into two subsets. Subset one will have L features, and subset two will have N - L features (with N being the number of features in the original dataset). The features are divided by generating random values from a uniform [0,1] distribution. If the random value is less than L/N, the feature is assigned to the first subset. If not, the feature is assigned to the second subset. (source: https://desktop.arcgis.com/en/arcmap/latest/extensions/geostatistical-analyst/how-subset-features-works.htm).

(4) We used the 10-fold cross-validation when training the models (Line 216 in the manuscript). The spatial distribution of the testing data is similar to the spatial distribution of all the data, which conforms with the stratified sampling scheme in machine learning.

(5) In order to find out if the model result is sensitive to the selection of weather stations used in the training and testing dataset, we conducted some experiments by randomly splitting the data into training and testing sets 50 times. We used the data from 1990, 2000, and 2010 to do the case study. As shown in Figure 1, the RMSE varies slightly from different scenarios of the test/training dataset, while there is no obvious variation in $R^2$ (Figure 2). In our study, we split the stations into testing and training stations in ArcGIS, which has considered the spatial distribution of the weather stations.

We added the details of the data splitting in the revised manuscript (**Lines 101-103, 484-488**).

[Figure]

Figure 1 The RMSE using different testing and training datasets

[Figure]

Figure 2 The $R^2$ using different testing and training datasets

**Q3: Only static features are used as predictors which implies that a model must trained for each month (!) of the period under consideration. Thus, dynamic information on the atmospheric state can exclusively deduced from the optimization procedure on the predictand. It is strongly recommended to introduce dynamical data as a predictor variable instead. Besides, the chosen predictors have periods with neglectable correlation with respect to the target quantity and important features such as the ambient topography (is the meteorological station located in a valley) is absent (see, e.g., Sha et al., 2020).**

Response: Many thanks for your comments.

(1) In our study, the model was trained for each month. We described the model construction in Lines 213-217. The longitude, latitude and elevation are indeed static factors, but we construct the model for each month, respectively, which can reflect the changes in temperature from month to month.

(2) The remote sensing data such as NDVI, land use change and surface temperature are usually not available before 2000 since our data is from 1951 to 2020. Furthermore, the MODIS data are not available for each month from January 2000 to December 2020. As shown in Figure 3, the percentage of the available MODIS images is low in northeast China and southern areas. So, the remote sensing data are not appropriate for generating long-term temperature data in our study.

(3) Furthermore, there is inherent data inaccuracy in the remote sensing data itself, such as the land use data. As shown in the study of Sha et al. (2020), the orography, as represented by elevation fields, can help characterize the spatial heterogeneity of 2-m temperature. The meteorological processes are locally embedded with small-scale terrain features. The terrain features including plain, slope, peak and valley are recognized as the semantic contents of terrain. They used the elevation data to represent those terrain semantics. In our study, we used the DEM data as the predictor in the machine learning models. As said in the study of Sha et al. (2020), the terrain semantics can be learned from gridded elevation inputs.

(4) We can obtain the high-resolution dataset using the selected predictors in our study. The accuracy evaluation shows the plausibility of the predictors. The model is robust in generating the long-term temperature datasets.

We discussed this in the Limitation section (**Lines 463, 477-483**).

[Figure]

Figure 3 Spatial distribution of the percentage of the available MODIS images in each year

(2000 - 2020) by excluding clouds.

**Q4: The evaluation does not serve the objectives of the study. The stations in the test dataset are dominated by stations over flat terrain with a dense observational network. Thus, potential deficiencies in capturing the variations due to underlying complex topography are hidden. Indeed, Figure 5 indicates that residuals are considerably larger over the mountainous region.**

Response: We agree with the comment. The meteorological stations in mainland China are unevenly distributed with more stations in the flat terrain and fewer stations in the mountains. This is the inherent data limitation for modelling continuous raster products in China (Guo et al., 2020; Liu et al., 2018). It is true that the stations in the Qinghai-Tibet plateau are sparse (Xu et al., 2018; Zhang et al., 2016). It is also an existing challenge of the spatial interpolation of temperature using the station data. The potential deficiency in capturing the variations in regions with complex terrain is an existing issue in the current studies. We are working to improve the accuracy of models in complex regions. The altitude information is conducive to the estimation of temperature (Berndt and Haberlandt, 2018).

To show the strength of our data in regions with complicated topography. We took the Tibetan plateau region as an example. We compared the accuracy of our data with Peng's data (Peng et al., 2019) in the Tibetan plateau. In our study, the accuracy in the Qinghai-Tibet Plateau is relatively good. We used the mean temperature data of Peng et al. (Peng et al., 2019) to make a comparison. The testing stations which were not used in the model training were used to make the comparison. As shown in Figure 4, the RMSE of most months for GPR is lower than Peng and GPR has a smaller variation in RMSE. The $R^2$ of GPR shows good accuracy from January to December, with smaller variation in each month, while for Peng's data the variation in summer is quite high.

We added it to the revised manuscript (**Lines 415-416**).

[Figure]

Figure 4 Comparison between the GPR data in our study and the Peng's data in the Tibetan Plateau

**Q5: Several issues in the follow-up study are present such as (a) a focus on large-scale temperature patterns instead of fine-scale patterns in Section 4.2. to reason the high spatial resolution of the dataset, (b) the interpretation of patterns in the Xinjiang region which look like artefacts (bulls-eye pattern in winter months) and (c) the missing notification on the better performance of the reference method ANUSPLIN for July-months in the 70s, 80s and 90s.**

Response: The spatial resolution of the temperature dataset in our study is 1 km. Since the territory of China is large, it is challenging to produce fine-scale temperature data. Besides, the scale of 1 km is the resolution of a lot of high-resolution datasets for mainland China, like the 1 km monthly temperature and precipitation dataset (Peng et al., 2019), 1 km daily surface air temperature product over mainland China (Chen et al., 2021), a high-resolution crop phenological dataset for three staple crops in China (Luo et al., 2020). The 1-km resolution is high enough for mainland China which can satisfy a lot of the requirements in other scientific research or practical applications. The 5-km spatial resolution dataset for Spain which is way smaller than China is also treated as the high-resolution dataset (Serrano-Notivoli et al., 2019). We admit that the finer resolution data may provide more detailed information but the 1-km resolution data is high enough for multiple studies.

(b) The bulls-eye pattern in the Xinjiang region in the winter months is induced by the complex topography, which just shows that the model captured the detailed local differentiation of temperature due to topographic conditions. As shown in Figure 5, the region which has a relatively lower temperature (Section 4.2 Figure 6) is because of the high altitude.

(c) Considering the proven power of ANUSPLIN in predicting meteorological variables, the GPR yields relatively satisfactory results. The accuracy of ANUSPLIN for July-months in the 70s, 80s and 90s is slightly higher than GPR while GPR still performs relatively well. ANUSPLIN uses the thin-plate smoothing spline algorithm which allows the introduction of multivariate linear sub-models with complex model coefficients to be calculated. Using the same computational resources, ANUSPLIN is more time-consuming than GPR. Besides, GPR has higher accuracy in the winter months.

We added the missing notification on the better performance of the ANUSPLIN for July-months in the 70s, 80s and 90s in the revised manuscript (**Lines 374-375**).

[Figure]

Figure 5 The elevation of Xinjiang Uygur Autonomous Region

**Q6: The comparison to the competing datasets ERA5 and FLADS is misleading due to the much coarser spatial resolution of these two datasets. A fair comparison would consult datasets with similar spatial resolution such as the dataset described in Peng et al., 2019.**

Response: Many thanks for your comments. We use three datasets: ERA5, FLDAS and TerraClimate. The spatial resolution of the three datasets is 27830 meters, 11132 meters, and 4638.3 meters, respectively. Our datasets are 1-km. We resampled all the data to 27830 meters to keep the resolution consistent and then we made comparisons. As shown in Figure 6, GPR still outperformed other products.

We replaced Figure 11 in the revised manuscript and added more text to make it clear (Lines **434-435**).

[Figure]

Figure 6 Taylor diagrams displaying a statistical comparison with observations between our products generated using the GPR model and the other products under the same spatial resolution.

Besides, we also compared our datasets with Peng's data as you suggested. As shown in Figure 7, our datasets have relatively higher accuracy than Peng's data on the whole, especially in warm months.

We also added this part in the Discussion section of the revised manuscript (**Lines 449-451**).

[Figure]

Figure 7 Accuracy comparison between the GPR data and the Peng's data for mean temperature

**Further minor issues are:**

**Q7: * Splitting into three distinct datasets is unnecessary. Rather merge it to one dataset with one DOI.**

Response: The Zenodo database has limitations for the data size (max 50 GB per dataset). The zip format file of each dataset is about 30 GB, so we uploaded mean, maximum, and minimum temperature data, respectively.

**Q8: * Refer to statistical and dynamical downscaling techniques in the introduction.**

Response: Thanks for your suggestion. The downscaling technique uses the existing coarse product to produce the high-resolution dataset. The interpolation uses the meteorological station data to generate the spatial continuous grid dataset. The two strategies use different source data, but they have the same objectives. In fact, there are multiple low spatial resolution datasets, such as the Climatic Research Unit (CRU) (Harris et al., 2014), the Global Precipitation Climatology Centre (GPCC) (Schneider et al., 2014; Becker et al., 2013), and Willmott & Matsuura (W&M) (Matsuura and Willmott, 2012) which are generated using the data from the observational stations. It is a reliable way to produce continuous datasets using the observed station data (Peng et al., 2019). We referred to the downscaling techniques in the introduction as suggested (**Lines 44-49**).

**Q9: \* Provide references to the problems related to remote sensing data (see I.66).**

Response: Thanks for your reminder. The references are listed below:

(Dong and Xiao, 2016)
(Xiao et al., 2018, p.2013–2016)
(Mao et al., 2019)

We provided the references in the revised manuscript (**Lines 79-80**).

**Q10: \* Describe the remapping of the STRM DEM data onto the 1x1 km grid (should be an averaging method).**

Response: We used GEE to export the STRM DEM data as a 1\*1 km grid. The "Scale" parameter was used to specify the output resolution to 1 km. The concept of "Scale" is illustrated in Figure 8 (https://developers.google.com/earth-engine/guides/scale). The default method of resampling is the nearest neighbour (https://developers.google.com/earth-engine/guides/scale#image-pyramids).

We clarified the resampling method in the revised manuscript (**Line 116**).

[Figure]

Figure 8 A graphic representation of an image dataset in Earth Engine. Dashed lines represent the pyramiding policy for aggregating 2x2 blocks of 4 pixels. Earth Engine uses the scale specified by the output to determine the appropriate level of the image pyramid to use as input.

**Q11: \* The used software tool MATLAB should be only mentioned once rather than being repeated three times. More details on the respective ML-technique would be appreciated.**

Response: Thanks a lot for your advice. We should mention MATLAB once. In the light of the limitation of the words in the manuscript, we did not provide so many details for all the machine learning methods but we provided the references or related links which have detailed descriptions of the machine learning methods.

We added some more information about the machine learning methods in the revised manuscript (**Lines 178, 192-195, 204**) and we deleted the redundant "MATLAB" as suggested.

**Q12: * l.149: Should be 'ensemble machine learning'**

Response: You are right. Thanks for pointing this out and sorry for the wrong spelling. We corrected it in the revised manuscript (**Line 179**).

**Q13: * l.219: "Unnecessary reference to Equation 4 which directly follows the sentence.**

Response: Thanks for your comment. The reference has been removed in the revised manuscript (**Line 242**).

**Q14: * l.343f. This is sentence is barely comprehensible.**

Response: We have revised the sentence in the revised manuscript (**Line 368**).


**Reviewer 2**

The comments offered have been immensely helpful. We appreciate your insightful comments on our paper. We have responded to every question, indicating exactly how we addressed each concern.

This study describes a new 1km dataset of monthly-mean, monthly-maximum and monthly-minimum surface temperature's over China, developed using machine learning methods. The method used for the final data set was chosen as the best performing method, after a comparison of three modern techniques. A dataset of 613 weather stations over China was used to train and test the machine learning methods. This study is very clearly written and the Figures are of high quality. I agree with all of reviewer 1's comments, so will not repeat these points and assume they have been addressed within the manuscript, but I will add a few further comments below.

Response: Many thanks for the constructive comments.

**General comments:**

**Q1: There is always a tradeoff between spatial and temporal resolution when designing new data products. Can you explain in a few sentences in the manuscript why you chose to create a product with such high spatial resolution but such low temporal resolution? You make comparisons at the end to the ERA5 dataset which does have much lower spatial resolution (~30 x less) but it has hourly temporal resolution (720 x more) which is very useful for a number of applications. Comments suggesting the applications where you think this dataset may be preferable to the others mentioned would also be useful.**

Response: Thanks a lot for your suggestion. Here are the reasons we produce the monthly product with high spatial resolution. First, the monthly temperature data is crucial for multiple studies and applications such as agriculture (Meshram et al., 2020), meteorological disasters (Tigkas et al., 2019) and ecology (Leihy et al., 2018). Second, the station data we obtained are from the China Meteorological Data Service Centre where the daily temperature data are not available. Thirdly, the daily temperature data with a high spatial resolution for a long period is enormously huge. Creating the data and storing the data for us is still quite challenging. ERA5 has high temporal resolution while the spatial resolution is low.

We commented on the applications of the monthly data in the revised manuscript (**Lines**

**39-40**).

**Q2: You mention ERA5 is only available from 1979, but it is now available back to 1950, so could be used to incorporate dynamical variables (as suggested by reviewer 1). I'm not suggesting you do this, but in the limitations this could be a point for future development. And the text should be updated to reflect the availability of ERA5.**

Response: The ERA5 is collected and processed on the Google Earth Engine platform, where the data is only available from 1979.

We revised the text to make it more accurate in the revised manuscript (**Line 135**).

**Q3: Is any quality control performed on the meteorological station data you use as inputs? A few stations with low quality data could skew the results in data sparse regions.**

Response: Yes. We train the model for each month. In each month, we deleted the stations with 999999 or 999998 values which means the station in that month has no data. We also checked the value range of the air temperature in different months and all the selected stations are with reasonable values.

We described the quality control of the data in the revised manuscript (**Lines 107-108**).

**Q4: Do you know if the final model output is sensitive to the choice of stations used in the test/training dataset? I imagine that this could heavily influence the results in the data sparse regions.**

Response: In order to find out if the model result is sensitive to the selection of weather stations used in the training and testing dataset, we conducted some experiments by randomly splitting the data into training and testing sets 50 times. We used the data from 1990, 2000, and 2010 to do the case study. As shown in Figure 9, the RMSE varies slightly from different scenarios of the test/training dataset, while there is no obvious variation in $R^2$ (Figure 10). In our study, we split the stations into testing and training stations in ArcGIS, which has considered the spatial distribution of the weather stations.

We discussed this in the revised manuscript in the Discussion session (**Lines 484-488**).

[Figure]

Figure 9 The RMSE using different testing and training datasets

[Figure]

Figure 10 The $R^2$ using different testing and training datasets

**Q5: Although you've included elevation, latitude and longitude there are multiple climatic regions in China, and a large amount of external drivers to variations in temperatures. The strength of these may modulate surface temperature behavior (e.g. the strength/location of the monsoon criculation, El Nino southern Oscillation, and other global teleconnections). Distance from the ocean could also play a role. Have you considered these in your explanations for months/stations with particulary large residuals, or stations with strange behaviors? It could be that if a month had anomalous large scale weather conditions, which your machine learning methods are not trained to capture there are large residuals? These could make interesting**

**case studies and could motivate future work incorporating some dynamical predictors.**

Response: Many thanks for your comments. The global teleconnections can influence surface temperature. This is a good topic for further study. We may use the data we generated combined with global teleconnections to do some research. In order to find out if the generated data in our study can capture the anomalous event, we did a case study in the Sichuan province. In 2006, there is an extremely severe drought event with an extremely high temperature in Sichuan (Li et al., 2011c). We extracted our mean temperature data to the stations which were not used in the model training. We used nine stations in Sichuan and compared the mean temperature in July from 2003 to 2009. As shown in Figure 11, the temperature in 2006 is markedly higher than the neighbouring years, which means that our data can capture the anomalous condition.

We have described the ability of our data to capture the anomalous condition in the revised manuscript (**Lines 429-430**).

[Figure]

Figure 11 Mean temperature for 9 different testing stations in Sichuan province

**Q6: Figure 2: This is a nice depiction of the relationships. If you could briefly unpack the meteorological understanding behind this in the text it would be beneficial to readers. Have you checked that the relationships hold if different climatic regions of China are subset out?**

Response: We used the subregions provided by Zhang et al. (Zhang et al., 2021). The subregions are divided according to the gradients of elevations and the precipitation patterns (Chen and Li, 2016; Zhang et al., 2021). The subregions are shown in Figure 12. We recalculated the correlation coefficients in each sub-region (Figure 13). The results show that Region 4, Region 5, Region 7, and Region 8 have clearly similar relationships as shown in Figure 2 in the manuscript. Since we train the model for the whole area of mainland China, the correlation in different sub-regions would not influence greatly on the whole region. It is an interesting topic to discuss different sub-regions, which can be helpful for regional studies. We will do the regional research in future work.

We added some text in the revised manuscript (**Lines 155-156**).

[Figure]

Figure 12 The division of the subregions, and the spatial distribution of the weather stations over the China mainland.

[Figure]

Figure 13 Correlation coefficients in different subregions

**Q7: Line 190-195. So you have 840 different models. Can you comment on how different are all the 70 models for each month? (e.g. do all the January models look very similar?) This could be useful to understand if there are dynamical meteorological explanations for any outliers.**

Response: We use the GPR model for all the 840 models for each month. The explicit basis in the GPR model is "constant" and the kernel function of the GPR algorithm is the exponential kernel. The predictor variables were standardized in the GPR. For each month, we use the temperature data of this month to train the model and then use this model to generate the grid data.

We added more details in the revised manuscript (**Lines 215-216**).

As the answer to Question 5: In order to find out if the generated data in our study can

capture the anomalous event, we did a case study in the Sichuan province. In 2006, there is an extremely severe drought event with an extreme high temperature in Sichuan (Li et al., 2011c). We extracted our mean temperature data to the stations which were not used in the model training. We used nine stations in Sichuan and compared the mean temperature in July from 2003 to 2009. the temperature in 2006 is markedly higher than the neighbouring years (Figure 14), which means that our data can capture the anomalous condition and it can be useful to understand the dynamical meteorological explanations for outliers.

[Figure]

Figure 14 Maximum temperature for 9 different testing stations in Sichuan province

**Q8: Line 243: Do you have a sense of why the errors are larger in the colder months? Are the impacts of local meteorological conditions larger in the cold season, which would make it more difficult for the methods to work? There may be meteorological literature on this.**

Response: Thanks for your comments. As shown in Figure 2 in the manuscript, there are two variables (i.e., longitude and elevation) that have higher correlation with temperature, while in cold months there is only one variable with relatively higher correlation (i.e., latitude). We did some literature review. The large-scale mountain area is an important source of uncertainty in the temperature mapping, which can influence the spatial distribution of the surface air temperature such as in the area of the Qinghai Tibet Plateau

and northwest China (Xu et al., 2018). The study of Stahl et al (Stahl et al., 2006) shows that the standard deviation for daily air temperature in winter is larger than that in summer. The study of Brunetti shows that the interpolation of high-resolution temperature for Italy has the lowest errors in spring and autumn and the highest errors in winter and explained that the elevation coefficients (lapse rates) are markedly different during winter (Brunetti et al., 2014). The air temperature in winter changes rapidly which may be a reason for the high estimation errors in winter (Amini et al., 2019). Rolland (Rolland, 2003) also found higher interpolation reliability for maximum and minimum temperature in summer than in winter.

We added the literature in the revised manuscript (**Lines 369-371**).

**Q9: Figure 6: Can you comment on any features you're resolving here that are not seen in the lower resolution gridded products you will compare to? There are some very high resolution features on the map, but clarification that they are physical would be useful.**

Response: The low spatial resolution can limit the ability to reflect the effects of complex topographies, land surface characteristics, and other processes on climate systems (Peng et al., 2019; Xu et al., 2017). The fine-scale data can provide realistic and reliable climate change information. We displayed the mean temperature of July, 2010 in the same region using ERA, FLADS datasets and the GPR dataset generated in our study (Figure 15). The GPR data can provide more spatial details than ERA and FLDAS (Figure 15).

We added the comment in the revised manuscript (**Lines 451-452**).

[Figure]

Figure 15 Comparison between the ERA, FLDAS and GPR datasets using the mean air temperature in July 2010

**Q10: Does your trend analysis agree with the existing literature on global warming over China? If so include references to this.**

Response: Thanks for your suggestion. We did a literature review and found some references which agree with the trend analysis in our study. The distribution of the temperature trend in China in our study agrees with the existing literature (Dong et al., 2015, p.1963–2012; Sun et al., 2018; You et al., 2021; Cui et al., 2017, p.1960–2015).

We added the literature in the revised manuscript (**Lines 342-343**).

**Q11: Figure 11: The Taylor diagrams show clear improvement from your new dataset. Also including some timeseries from locations not sampled from the observation network compared between the three datasets would be useful to understand how the four products sample the seasonal cycles of the variables.**

Response: Many thanks for your advice. We randomly selected three locations not sampled from the observation network to make a comparison. The location of the points is shown in Figure 16. We extract the ERA, FLDAS and the GPR mean air temperature data to the three new points to make a comparison. As shown in Figure 17, the mean air temperature data of the three datasets have a similar pattern from January 1982 to December 2019. We do not have actual temperature data from the three randomly selected points, so it is not possible to make an accuracy comparison. However, the ERA and FLADS datasets are already widely used in a lot of studies (McNally et al., 2017; Hersbach et al., 2020), which means they are reliable to some extent. Thus the similar pattern between the GPR data and the ERA/FLDAS data can show the reliability of the GPR dataset to a certain extent.

[Figure]

Figure 16 The location of the three random points

[Figure]

Figure 17 Comparisons of the data time series for mean air temperature in three random

points from January 1982 to December 2019

**Small corrections:**

**Q12: The height of the air temperatures (surface, 1.5m, 2m) should be added to the manuscript when this is mentioned.**

Response: The height of the air temperatures from the weather stations is 2 m.

We added this to section 2.1 Meteorological station data in the revised manuscript (**Lines 99-100**).

**Q13: The acronyms for datasets/methods should be defined in the abstract to make it easier to read.**

Response: ANUSPLIN: (short for Australian National University Spline); TerraClimate: Monthly Climate and Climatic Water Balance for Global Terrestrial Surfaces; FLDAS: Famine Early Warning Systems Network (FEWS NET) Land Data Assimilation System, and ERA5: ECMWF Climate Reanalysis.

We added the full definition in the Abstract in the revised manuscript (**Lines 19, 22-23**).

**Q14: Line 38: after commenting on the limitations of the observing stations you could comment here on the limitations of reanalysis based products.**

Response: Thanks for the suggestion. The reanalysis based data usually have a low spatial resolution, which limits their ability to reflect the effects of complex topographies, land surface characteristics, and other processes on climate systems (Peng et al., 2019; Xu et al., 2017).

We added this comment to our revised manuscript (**Lines 49-51**).

**Q15: Throughout the text when you say 'high resolution' this should be changed to 'high spatial resolution' e.g. line 55.**

Response: Yes. It has been changed to "high spatial resolution" in the revised manuscript (**Line 66**).

**Q16: Line 56: 'traditional interpolation techniques' might be clearer?**

Response: We agree with this. We changed it in the revised manuscript (**Line 67**).

**Q17: Line 58-60: You comment on a few studies which talk about the superior performance of machine learning techniques but you do not say what the**

**benchmark is that they've succeeded against. This should be included.**

Response: The machine learning methods were selected mainly according to the applications of machine learning methods in previous studies. There is potential in applying machine learning methods to predict spatially continuous variables. The combination of machine learning and the traditional model can usually have better performance (Appelhans et al., 2015; Li et al., 2011b). Secondary information considered such as slope, latitude and longitude can improve the performance of machine learning as they provide essential information for machine learning methods (Li et al., 2011b; Alizamir et al., 2020; Appelhans et al., 2015; Kisi et al., 2017; Zhu et al., 2018).

We included additional descriptions in the revised manuscript (**Lines 71-72**).

**Q18: Line 61: 'estimation of short-term air temperature ' – I'm not sure what you mean by this?**

Response: We are sorry for making you confused. Here we deleted "short-term" to make it clearer (**Line 74** in the revised manuscript).

**Q19: Line 86: The link here gives me an Error 404.**

Response: The link is not available recently, because the data are not available online anymore. We changed our link to the homepage (https://data.cma.cn/data/). See **Line 99**.

**Q20: Around the discussion for Figure 1 it would be interesting to know the spatial distance between observation sites. This might be a small indication of confidence in the final machine learning model output.**

Response: Thanks for your advice. We generated the Euclidean distance for the observation stations. Figure 18 shows that the Euclidean distance is quite small in most regions. The Euclidean distance is relatively large in the west of the Tibetan Plateau and the small region in Inner Mongolia. The larger Euclidean distance means the stations in that region are sparse, which can have an impact on the interpolation accuracy (Hijmans et al., 2005; Li et al., 2011a).

We talked about the model output and the spatial distance between observation sites in the revised manuscript (**Lines 471-473**).

[Figure]

Figure 18 Euclidean distance of the weather stations

**Q21: Section 2.3: When commenting on the spatial resolution of the gridded products used for comparison it would be useful to also have this in km.**

Response: Thanks for your suggestion. The spatial resolution of TerraClimate, FLDAS, and ERA5 are about 4.6 km, 11 km and 28 km, respectively.

We updated the text in the revised manuscript (**Lines 121, 124, 127**).

**Q22: Line 149: 'machining learning' should be 'machine learning'**

Response: We changed the wrong spelling (**Line 169** in the revised manuscript).

**Q23: Section 3.2.2 Are the choices of parameters for the SVM method standard in the literature? Can you please comment on your choices?**

Response: Thanks for your suggestion. For the SVM models, we used the Gaussian kernel. The kernel scale is set to 1.7 for all SVM models. Hyperparameters are important for the performance of the model. We optimized the kernel scale of SVM and made a comparison. The box constraint and the epsilon hyperparameters are varying from month to month according to the training data of each month. The mean temperature data from January to December from 1951 to 2020 was used for comparing the new model with varying kernel scales and the used model in our study. The accuracy of each month between the optimized model and the used model in the paper is similar (Figure 19), the optimized models do not improve significantly. As we can see that the adjustment of the hyperparameters has little impact on the model accuracy in our study. Besides, the models

used in our study are more time-saving and efficient.

We added more information in the revised manuscript (**Lines 192-193, 194-195**).

[Figure]

Figure 19 Comparison between optimized SVM model (new) and the used model in the paper (old).

**Q24: Line 342: ' shows a cyclic pattern' might be clearer.**

Response: We changed the text as suggested in the revised manuscript (**Line 367**).


**Reviewer 3**

The comments offered have been immensely helpful. We appreciate your insightful comments on our paper. We have responded to every question, indicating exactly how we addressed each concern.

**General Comments:**

The paper is presenting a method for spatial interpolation of air temperature data for China from meteorological stations based on machine learning tools. The authors analyze the technique used and present the limitations of the experiment. Three ML models were tested and three interpolation method and the Gaussian Process Regression was chosen based on its better performance. The results compared with existing published datasets. A detailed trend analysis of the predicted dataset is also presented. ML techniques are very promising as they are addressing current challenges in computational research.

Response: Many thanks for the constructive comments on our study.

**Specific comments:**

**Q1: It is not clear to me if all stations contribute equally to the analysis, for example red and blue stations (Figure S2). Is there any weighting technique applied to the training model(s)? If yes, I think it could be mentioned.**

Response: In our study, we used the "subset features" option of the Geostatistical Analyst Tools in ArcGIS10.8 to divide the original dataset into 70% training dataset and 30% testing dataset. This tool considers the randomness both in the data and the spatial distribution of the data as described in the manuscript (**Line 101-104**). There is no weighting technique applied in the training models.

**Q3: Besides the trends statistical analysis, are there available error spatial distributions of the predicted temperatures so as to illustrate the confidence level of the analysis results especially in station empty areas?**

Response: GPR is a full Bayesian learning algorithm. A process is referred to as a Gaussian process if it is assumed that the joint probability distribution of model outputs is Gaussian (Zhu et al., 2018). Because a GPR model is probabilistic, it is possible to compute the prediction intervals using the trained model. Here, we displayed the spatial

distribution of the width of the predicted intervals with a significance level of 5% (using the upper limit minus the lower limit) for the 12 months in 2010 using the trained models (Figure 20). Figure 21 provides the histogram of the width of the confidence intervals. Most of the regions have small confidence intervals. The calculation was conducted in Matlab 2021b. For more detailed information about the prediction intervals of GPR models, please see https://www.mathworks.com/help/stats/gaussian-process-regression-models.html.

We added the uncertainty discussion in the revised manuscript (**Lines 203-205, 473-476**).

[Figure]

Figure 20 The spatial distribution of the width of the 95% prediction intervals (the upper limit minus the lower limit of the confidence interval) for 12 months in 2010

[Figure]

Figure 21 The histogram of the width of the 95% prediction intervals (the upper limit minus the lower limit of the confidence interval) for 12 months in 2010

**Q4: In addition, as the study of the climatic dynamics is in the epicenter of this work, the use of remote sensing data jointly with the land meteo stations could overcome the data scarcity, improve the results and reveal trends with more accuracy after 2000.**

Response: The goal of our study is to generate the long-term time series of high-resolution temperature data. Due to the limitation of the remote sensing data, we did not consider remote sensing data in our study. We discussed the topic in the discussion session (see **Lines 477-483**).

The longitude, latitude and elevation are static factors, but we construct the model for each month, respectively, which can reflect the changes of temperature from month to month. The remote sensing data such as NDVI, land-use change and surface temperature are usually not available before 2000 since our data is from 1951 to 2020. Furthermore, the MODIS data are not available for each month from January 2000 to December 2020. As shown in Figure 3, the percentage of the available MODIS images are low in northeast China and southern areas. So, we did not use the remote sensing data for generating long-term temperature data in our study. We will consider using the remote sensing data in future studies to further increase the accuracy.

[Figure]

Figure 22 Spatial distribution of the percentage of the available MODIS images in each year (2000 - 2020) by excluding clouds.

**Q5: While the height of the air Temperature is mentioned for the ERA5 dataset (2m), this is not the case for the GPRChinaTemp1km product or the other datasets**

**mentioned in the analysis.**

Response: We will include more detailed descriptions of the data we used. In our study, we use the weather station data to interpolate the gridded temperature datasets. The station data used in the study records the temperature at 2 m height above ground (Liu et al., 2011; Zhang et al., 2010).

We mentioned this in the revised manuscript (**Lines 99, 436-437**). Thus, the generated GPRChinaTemp1km product also represents the temperature data at 2 m height.

For TerraClimate data, it is produced based on other datasets including WorldClim, CRUTs4.0 and JRA-55 (Abatzoglou et al., 2018, p.1958–2015). The temperatures in WorldClim are at 2 m height (Fick and Hijmans, 2017; Chou et al., 2020). The temperature data from CRU Ts and JRA-55 are also at 2 m height (Harris et al., 2020; https://jra.kishou.go.jp/JRA-55/document/JRA-55_handbook_LL125_en.pdf). Therefore, the TerraClimate dataset also represents the 2m temperature.

The height of the temperature data from FLDAS is also 2 m (McNally et al., 2017; https://ldas.gsfc.nasa.gov/fldas/specifications).

We included the detailed information on the height of the data in the revised manuscript (**Lines 137-141, 436-437**).

**Q6: Could the experiment be tested in other atmospheric parameters? If so, I think that a few sentences on the perspectives of the specific approach would be beneficial.**

Response: Thanks a lot for your suggestion. Our current study is only aiming for temperature. We have not done some experiments on other meteorological variables. This actually is our next work. We are trying to apply the GPR model to other meteorological variables.

We added a few sentences in the Discussion session (**Lines 430-431**).

---

## Author Response (AR2)

Dear Martin Schultz,

Thank you for your letter and the chance of revising our paper on "GPRChinaTemp1km: a high-resolution monthly air temperature dataset for China (1951–2020) based on machine learning" (Manuscript ID: essd-2021-442). We thank you a lot for giving our manuscript insightful comments to further improve our manuscript.

We have revised our manuscript following your advice. We have included the comments in this letter and responded to them individually. The revisions have been approved by all three authors. The responses to the comments are listed below in blue.

thank you for carefully considering the reviewer comments and submitting a revised manuscript version. While most comments have been resolved, there are two important issues that remain and which need to be addressed through further analysis, before the paper can be published:

(1) please remove the sentence "This tool considers the randomness both in the data and the spatial distribution of the data", because this statement is misleading. The reviewers' point is that the data are *not* randomly distributed but correlated in space. You can obtain a more robust estimate of the model error by cutting out a region of, say, 300-400 km length in longitude and latitude and using all stations inside this region for testing, while stations outside this rectangle are used for training. By randomly positioning this rectangle a few times, you will get a useful uncertainty measure.

Response: Thanks a lot for your constructive comment. We deleted the sentence you mentioned in the revised manuscript as requested (Lines 103-104).

As you suggested, we conducted an experiment using rectangles. We cut out the study area region using squares with the side length of 300 km in longitude and latitude (Figure 1).

The stations inside the square were used as the testing stations and the stations outside the square were used for training the models. Sixteen years of data were used for the experiment: 1980, 1983, 1985, 1988, 1990, 1993, 1995, 1998, 2000, 2003, 2005, 2008, 2010, 2013, 2015, and 2018 (We selected some years to do the experiment due to the huge data volume and the time-consuming modelling running on our computer). For the data of each month per year, the square numbers are 142. We trained models for each month per year using the training stations outside the squares 142 times, so 27264 models were trained.

We obtained the accuracy (MAE, RMSE, and $R^2$) for the testing stations in each square. The accuracy results are shown in Figure 2. The MAE, RMSE and $R^2$ (median values of RMSE, MAE and $R^2$ are 0.49, 0.56 and 0.92, respectively) indicate that the accuracy of the GPR models by cutting out a square as testing is high, which can prove the

robustness of the GPR models. The monthly comparison also shows the stability of the GPR models in different months using the square strategy (Figure 3).

[Figure]

Figure 1 Illustration of the squares used for the experiments.

[Figure]

Figure 2 Boxplots of the accuracy of the testing stations for the cut-out 30-km squares of all months in all years.

[Figure]

Figure 3 Boxplots of the accuracy of the testing stations for the cut-out 30-km squares for each month.

Besides, we also compared the accuracy of the testing stations using the rectangle strategy with the accuracy of the testing stations used in our original study. Figure 4 shows the scatter plots of the predicted mean temperature (Tmean) in our original study and the predicted Tmean obtained by the square strategy for the testing stations. It can be found that the predicted values by the square strategy are almost the same as the predicted values in our study, which can confirm the robustness of the models used in our study. We also provided all the scatter plots of each month of all years in a ZIP file. The accuracy comparison of MAE, RMSE and $R^2$ between the original prediction (i.e. the prediction in our original study) and the prediction obtained by the square strategy also shows that they have similar accuracy.

[Figure]

Figure 4 Scatter plots between the original predicted Tmean and the predicted Tmean using the square for the testing stations. The scatter plots for the rest months in different years are provided in a ZIP file.

[Figure]

Figure 5 The accuracy comparison between the original prediction and the new prediction using the square strategy.

We added this discussion in the revised manuscript (Lines 393-395).

(2) a more critical test of the model's interpolation capabilities can be obtained by evaluating spatial anomalies (i.e. monthly value of a specific year minus mean monthly value over all years). If the model has real power, then it should be able to reproduce such spatial anomaly patterns, for example during the heat episode in Sichuan in 2006, which you showed. On the other hand, if longitude, latitude and altitude are the only predictor variables, one might expect that the spatial anomalies look identical each year.

Response: Thanks a lot for your valuable advice. As you requested, we did a test of the model's interpolation capabilities. We obtained the spatial anomalies for each month over all years. We collected some historical heat/cold wave events as well as some droughts which are related to heat waves as the events to validate the applicability of our datasets. The events are mainly selected based on previous literature and some Chinese news, as well as the historical documents in our research group; the references have been put in the caption of the figures. Eleven events in total are used for the validation. We presented the spatial mean temperature anomaly of the year with the extreme temperature-related events as well as the mean temperature anomaly of the two years before and two years after the event year.

The results are shown in Figures 6-16, and the event year has a different colour of the frame. It can be found that our datasets can reproduce the spatial anomaly patterns in the year with extreme temperatures. Besides, the spatial anomalies for the same month in different years are different because we trained the model for each month using the monthly temperature in that month which consider the dynamic characteristics of the

temperature. Just as shown in Figure 66-16, the spatial difference can be clearly seen in the same month for different years.

We have added this part in the Discussion session in our revised manuscript (Lines 433-434). We also added these spatial maps in our Supplementary materials (Figures S60-70)

[Figure]

Figure 6 Comparison between the year with higher mean temperature anomaly (for Tmean) and the adjacent years. A drought event happened in Sichuan and Chongqing and neighbouring regions in July 2006 which is associated with heat waves (Li et al., 2011).

[Figure]

Figure 7 Comparison between the year with lower mean temperature anomaly (for Tmean) and the

adjacent years. Hubei, Anhui, Hunan and Jiangxi and surrounding areas were affected by cold temperatures in the 2008 Chinese winter storms (Liu et al., 2016; Zhou et al., 2014; Lu et al., 2010).

[Figure]

Figure 8 Comparison between the year with higher mean temperature anomaly (for Tmean) and the adjacent years. Heat waves hit Shanghai and neighbouring regions in July 2013 (Pu et al., 2017; Ding and Ke, 2015; Li et al., 2015; Jing-Bei, 2014).

[Figure]

Figure 9 Comparison between the year with extremely high temperature and the adjacent years in Shandong. A drought event with heat wave happened in 1988 in Shandong province and neighbouring areas.

[Figure]

Figure 10 Comparison between the year with higher mean temperature anomaly (for Tmean) and the adjacent years. Heat waves happened in Beijing and the surrounding areas in August 1994 (Zhang et al., 2018; Park et al., 2012).

[Figure]

Figure 11 Comparison between the year with higher mean temperature anomaly (for Tmean) and the adjacent years. Heat waves hit Beijing and the neighbouring regions in July 1997 (Park et al., 2012; Zhang et al., 2018).

[Figure]

Figure 12 Comparison between the year with lower mean temperature anomaly (for Tmean) and the adjacent years. Cold waves hit the Tibetan Plateau and neighbouring areas in September 1997.

[Figure]

Figure 13 Comparison between the year with higher mean temperature anomaly (for Tmean) and the adjacent years. Drought happened in Hunan, Jiangxi, Zhejiang and Fujian provinces and neighbouring regions in July 2003 which is caused by the lack of precipitation and heat wave (Wang and Yan, 2021; Zhang et al., 2017; Ding and Ke, 2015). In 2001, the summer drought with high temperature happened in the southern China as well, such as Hubei, Zhejiang (Pandey et al., 2007, p.36–37). There were also heatwaves in the southern cities in 2005, for example Guangdong province (Yang et al., 2013).

[Figure]

Figure 14 Comparison between the year with lower mean temperature anomaly (for Tmean) and the adjacent years. Guizhou, Guangxi, Hubei, Hunan, Anhui, and Jiangxi as well as the neighbouring regions were affected by cold waves in January 2011 (Qi et al., 2017; http://www.gov.cn/jrzg/2011-01/05/content_1778886.htm, last access: 18 May 2022).

[Figure]

Figure 15 Comparison between the year with lower mean temperature anomaly (for Tmean) and the adjacent years. A cold wave happened in Inner Mongolia and neighbouring regions in January 2016 (Ma and Zhu, 2019; Jiang et al., 2018).

[Figure]

Figure 16 Comparison between the year with higher mean temperature anomaly (for Tmean) and the adjacent years. Extremely high temperatures occurred in Inner Mongolia and Qinghai and other neighbouring regions in August, 2016 (http://www.cma.gov.cn/2011xwzx/2011xqxxw/2011xqxyw/201609/t20160902_320919.html, last access: 18 May 2022).

**References**

Ding, T. and Ke, Z.: Characteristics and changes of regional wet and dry heat wave events in China during 1960–2013, Theor Appl Climatol, 122, 651–665, https://doi.org/10.1007/s00704-014-1322-9, 2015.

Jiang, D., Xiao, W., Wang, J., Wang, H., Zhao, Y., Li, B., and Zhou, P.: Evaluation of the effects of one cold wave on heating energy consumption in different regions of northern China, Energy, 142, 331–338, https://doi.org/10.1016/j.energy.2017.09.150, 2018.

Jing-Bei, P.: An Investigation of the Formation of the Heat Wave in Southern China in Summer 2013 and the Relevant Abnormal Subtropical High Activities, 7, 286–290, https://doi.org/10.3878/j.issn.1674-2834.13.0097, 2014.

Li, J., Ding, T., Jia, X., and Zhao, X.: Analysis on the Extreme Heat Wave over China around Yangtze River Region in the Summer of 2013 and Its Main Contributing Factors, 2015, e706713, https://doi.org/10.1155/2015/706713, 2015.

Li, Y., Xu, H., and Liu, D.: Features of the extremely severe drought in the east of Southwest China and anomalies of atmospheric circulation in summer 2006, Acta Meteorol Sin, 25, 176–187, https://doi.org/10.1007/s13351-011-0025-8, 2011.

Liu, Y., Wang, M., Wang, W., Fu, H., and Lu, C.: Chilling damage to mangrove mollusk species by

the 2008 cold event in Southern China, 7, e01312, https://doi.org/10.1002/ecs2.1312, 2016.

Lu, Q., Zhang, W., Zhang, P., Wu, X., Zhang, F., Liu, Z., and Dale, M. B.: Monitoring the 2008 cold surge and frozen disasters snowstorm in South China based on regional ATOVS data assimilation, Sci. China Earth Sci., 53, 1216–1228, https://doi.org/10.1007/s11430-010-3040-1, 2010.

Ma, S. and Zhu, C.: Extreme Cold Wave over East Asia in January 2016: A Possible Response to the Larger Internal Atmospheric Variability Induced by Arctic Warming, 32, 1203–1216, https://doi.org/10.1175/JCLI-D-18-0234.1, 2019.

Pandey, S., Bhandari, H. S., and Hardy, B.: Economic Costs of Drought and Rice Farmers' Coping Mechanisms: A Cross-country Comparative Analysis, Int. Rice Res. Inst., 36–37 pp., 2007.

Park, J.-K., Lu, R., Li, C., and Kim, E. B.: Interannual variation of tropical night frequency in Beijing and associated large-scale circulation background, Adv. Atmos. Sci., 29, 295–306, https://doi.org/10.1007/s00376-011-1141-1, 2012.

Pu, X., Wang, T. J., Huang, X., Melas, D., Zanis, P., Papanastasiou, D. K., and Poupkou, A.: Enhanced surface ozone during the heat wave of 2013 in Yangtze River Delta region, China, Science of The Total Environment, 603–604, 807–816, https://doi.org/10.1016/j.scitotenv.2017.03.056, 2017.

Qi L., Ma Q., and Zhang W.: Verification of forecasting capability of cold wave process in the winter of 2011 /2012 with GRAPES, 40, 791–802, https://doi.org/10.13878/j.cnki.dqkxxb.20150104001, 2017. (in Chinese)

Wang, J. and Yan, Z.: Rapid rises in the magnitude and risk of extreme regional heat wave events in China, Weather and Climate Extremes, 34, 100379, https://doi.org/10.1016/j.wace.2021.100379, 2021.

Yang, J., Liu, H. Z., Ou, C. Q., Lin, G. Z., Ding, Y., Zhou, Q., Shen, J. C., and Chen, P. Y.: Impact of Heat Wave in 2005 on Mortality in Guangzhou, China, Biomedical and Environmental Sciences, 26, 647–654, https://doi.org/10.3967/0895-3988.2013.08.003, 2013.

Zhang, S., Huang, G., Qi, Y., and Jia, G.: Impact of urbanization on summer rainfall in Beijing–Tianjin–Hebei metropolis under different climate backgrounds, Theor Appl Climatol, 133, 1093–1106, https://doi.org/10.1007/s00704-017-2225-3, 2018.

Zhang, Y., You, Q., Chen, C., and Li, X.: Flash droughts in a typical humid and subtropical basin: A case study in the Gan River Basin, China, Journal of Hydrology, 551, 162–176, https://doi.org/10.1016/j.jhydrol.2017.05.044, 2017.

Zhou, M. G., Wang, L. J., Liu, T., Zhang, Y. H., Lin, H. L., Luo, Y., Xiao, J. P., Zeng, W. L., Zhang, Y. W., Wang, X. F., Gu, X., Rutherford, S., Chu, C., and Ma, W. J.: Health impact of the 2008 cold spell on mortality in subtropical China: the climate and health impact national assessment study (CHINAs), Environmental Health, 13, 60, https://doi.org/10.1186/1476-069X-13-60, 2014.